# Hista and Numca: Estimate State Value Effectively for Large Language Model Reinforcement Learning

**Zizhe Chen**[1]  **Jiqian Dong**[2]  **Yizhou Tian**[1]  **Garry Yang**[1]  **Yongqiang Chen**[1]  **Zhitang Chen**[2]  **James Cheng**[1]

## Abstract

Reinforcement learning (RL) refines large language models (LLMs) by directly optimizing model behavior through reward signals. While accurate state value estimation is critical for stable training in classical RL, it remains an underexplored challenge in LLM post-training. In this work, we introduce the State Value Estimation Benchmark (SVEB) to assess state estimation within existing RL frameworks and show that critics in standard approaches like PPO collapse to a coarse group-average baseline. To address this, we propose two techniques: *Numca*, which leverages numerical spans as gradable milestones for state value estimation, and *Hista*, a framework that uses LLM's hidden states as representation to weighted average disjoint rollouts and their return. Extensive experiments demonstrate that both methods yield more accurate state value estimates and enhance training performance across different RL algorithms and model sizes without incurring significant computational overhead. Code available at https://github.com/VOXXXX1874/Hista.

## 1. Introduction

The effectiveness of DeepSeek-R1 established GRPO as a foundational RL post training method for LLMs (Shao et al., 2024) (Guo et al., 2025). Recent progress has focused on creating more advanced successors. These newer algorithms, such as DAPO (Yu et al., 2025), GSPO (Zheng et al., 2025), and CSIPO(MiniMax et al., 2025), surpass GRPO by optimizing components like KL-regularization, importance sampling calculation, and clip mechanism.

Despite these improvements, all these successors retain a

---

[1]Department of Computer Science and Engineering, The Chinese University of Hong Kong [2]Huawei Technologies Ltd. Correspondence to: James Cheng <jcheng@cse.cuhk.edu.hk>.

*Proceedings of the 43rd International Conference on Machine Learning*, Seoul, South Korea. PMLR 306, 2026. Copyright 2026 by the author(s).

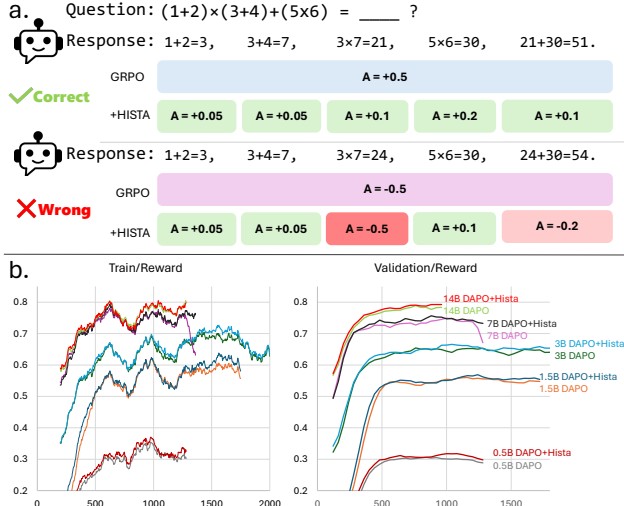

*Figure 1.* **a.** Credit assignment of *Hista*. **b.** Apply *Hista* to DAPO on Qwen2.5 base models, where each color represents one settings. Applying *Hista* achieves better convergence speed and reward.

core limitation inherited from GRPO: treating entire response as single action and apply the same state value to every token. This simplification stems directly from the difficulty of parsing intermediate states. More sophisticated approach, such as training process-level reward models (Lightman et al., 2023; Zhang et al., 2025) or applying Monte Carlo tree search (Li et al., 2025; Guan et al., 2025) attempt to mitigate this problem, but they inevitably incur prohibitive costs in labeled data or computational rollouts, rendering them impractical for large-scale training.

Given this reliance on coarse group-averaged reward baseline, current RL training methods for LLMs deviate strikingly from classical RL, whose central theme is the use of a learned state value function $V^\pi(s)$ to estimate the expected return from any given state (Konda & Tsitsiklis, 1999; Schulman et al., 2017). The critic network provides this fine-grained valuation, enabling stable and guided policy updates by the actor. This contrast raises a question:

*How does accurate state value improve LLM RL training, and how can it be estimated effectively?*

To answer this question, we first construct a State Value Estimation Benchmark (SVEB) to quantify the discrepancy

between ground-truth state values and those estimated by various methods. Using this benchmark, we uncover a key limitation of standard PPO in LLM RL: although PPO includes a critic network that ostensibly provides fine-grained token-level credit assignment, the state values it produces consistently collapse toward the **group-averaged reward**. In effect, this degrades the method to a GRPO-like baseline, offering no more informative guidance than a coarse group average. Empirically, we conduct extensive experiments across different data sources using PPO and find that its value estimates inevitably regress to this average.

To begin mitigating this issue, reframing mathematical problem solving as a sparse-reward goal-reaching problem. Inspired by Hindsight Experience Replay (HER) (Andrychowicz et al., 2018), which relabels past experiences with alternative goals, we seek to identify meaningful "milestones" within a text sequence. In mathematical reasoning (e.g., the tasks in AIME), we observe that numeric values naturally serve as easily parsable milestones. Based on this insight, we introduce *Numca*, a heuristic method that utlizes numbers as representation to group disjoint rollouts. *Numca* outperforms group-average reward both on our SVEB benchmark and in downstream RL training.

While *Numca* demonstrates the value of accurate state value, its reliance on domain-specific heuristics limits generality. To address this, we further propose *Hista*, a general framework that represents each generation step using its hidden state embeddings. *Hista* estimates state values via probability-weighted reward averaging, requiring no additional rollouts or training. This approach operationalizes the intuition that semantically similar states should receive similar state values. In Theorem 5.5, we prove that *Hista* yields more accurate estimates than the group-average baseline. Empirically, *Hista* outperforms baselines across all domains in SVEB, and RL-trained models equipped with *Hista* surpass strong alternatives such as DAPO and CSIPO. In summary, this work makes three primary contributions:

**SVEB**: We propose the SVEB for evaluating state value estimation methods. Using SVEB, we identify fundamental limitations of standard PPO in the LLM setting.

**Numca:** We introduce *Numca*, a simple and highly efficient heuristic that provides effective state value estimation for mathematical reasoning.

**Hista:** To build a general solution, we propose the *Hista* framework based on hidden states, with theoretical guarantee that *Hista* delivers better value estimates than GRPO.

## 2. Related Work

**Classic RL:** To achieve stable policy optimization, actor-critic methods combine value-based (Mnih et al., 2015) and policy-based (Sutton et al., 1999a) reinforcement learn-ing by utilizing an actor network for action generation and a critic network for value estimation, as exemplified by PPO (Schulman et al., 2017). Following PPO, extensive efforts have been devoted to improving critic accuracy and stability. TD3 mitigates critic approximation errors through clipped double-Q learning (Fujimoto et al., 2018), while subsequent works further explore conservative, uncertainty-aware, and entropy-regularized value estimation (Haarnoja et al., 2018; Kostrikov et al., 2021; Kumar et al., 2020). Another important direction is Goal-Conditioned RL (GCRL) (Schaul et al., 2015), which studies credit assignment under sparse rewards. HER addresses sparse goal-reaching tasks through hindsight goal relabeling (Andrychowicz et al., 2018), while more recent works focus on representation learning and reachability modeling for credit assignment in a more general settings (Eysenbach et al., 2023; Wang et al., 2023). Interestingly, this perspective is closely related to option discovery in Hierarchical Reinforcement Learning (Sutton et al., 1999b), where successor representations and latent state geometry are leveraged to discover reusable temporally-extended subgoals (Ramesh et al., 2019; Machado et al., 2023).

**Credit Assignment in LLM RL:** Finer-grained credit assignment is also a central problem in reinforcement learning for large language models. PPO-style methods such as VAPO and VC-PPO (Yue et al., 2025; Yuan et al., 2025) introduce value pretraining techniques to alleviate critic initialization mismatch, but they still require training a critic with a scale comparable to the actor. Another line of work leverages Monte Carlo sampling or tree search to obtain more accurate state-value or action-value estimation (Li et al., 2025; Guan et al., 2025; Abdin et al., 2024). Although these methods avoid maintaining a dedicated critic network, their computational cost grows rapidly with response length and sampling budget. Process Reward Models (PRMs) (Zhang et al., 2025; Yuan et al., 2024; Lightman et al., 2023) provide dense supervision over intermediate reasoning steps, but they mainly serve as verifiers of reasoning correctness rather than explicit estimators of state value for policy optimization. Several lightweight alternatives have also been proposed to improve credit assignment (Sun et al., 2025; Wang et al., 2025), though they are generally not derived from the perspective of state-value estimation. In contrast, our method directly exploits hidden-state representations to achieve computational efficiency comparable to GRPO while providing provably better state-value estimation.

## 3. Preliminary

### 3.1. MDP Modeling in LLM Generation

The autoregressive generation of LLMs is naturally formalized as a Markov Decision Process (MDP), $\mathcal{M} = (\mathcal{S}, \mathcal{A}, P, R, \gamma)$. Let $\mathcal{V}$ denote the vocabulary of discrete

tokens, and let $\langle \text{eos} \rangle \in \mathcal{V}$ be the end-of-sequence token. A state $s_t = (x_1, \ldots, x_t) \in \mathcal{S}$ is the sequence of tokens produced until time $t$, including system prompt, user prompt, and previously generated tokens. The initial state $s_0$ is the the prompt itself. The action space at any states is $\mathcal{A}(s_t) = \mathcal{V}$, where an action $a_t \in \mathcal{V}$ selects the next token. The transition is deterministic with $P(s_{t+1} \mid s_t, a_t) = 1$. A state is terminal if $a_t = \langle \text{eos} \rangle$ or the maximum length is reached. The discount factor $\gamma = 1$ and the reward is only provided upon termination:

$$R(s_t, a_t, s_{t+1}) = \begin{cases} r(s_{t+1}), & s_{t+1} \text{ terminal,} \\ 0, & \text{otherwise.} \end{cases}$$

## 3.2. Value Estimation in LLM Reasoning

Building upon the MDP, the state value $V^\pi(s_t)$ quantifies the expected cumulative future reward starting from that partial sequence:

$$V^\pi(s_t) = \mathbb{E}_\pi \left[ \sum_{k=t}^{T-1} R(s_k, a_k, s_{k+1}) \,\middle|\, s_t \right] = \mathbb{E}_\pi[r(s_T) \mid s_t],$$

In popular LLM RL frameworks such as TRL (von Werra et al., 2020), VeRL (Sheng et al., 2024), and OpenRLHF (Hu et al., 2024), the critic network $V_\phi$ is trained to provide token-level state value estimates in PPO. Given a generation rollout $\tau = (s_0, a_0, \ldots, s_T)$, the temporal-difference (TD) residual at time $t$ is defined as:

$$\delta_t = r_t + V_\phi(s_{t+1}) - V_\phi(s_t), \quad \text{with} \quad V_\phi(s_{T+1}) = 0$$

Advantages are then computed via Generalized Advantage Estimation (GAE) (Schulman et al., 2018): $\hat{A}_t = \sum_{i=0}^{T-t} (\lambda\gamma)^i \delta_{t+i}$, where $\lambda$ is a smoothing hyperparameter and $\gamma$ is the discount factor. The critic is trained by regressing its predictions toward an empirical target. The critic loss is formulated as the mean squared error between the predicted state value and the target value constructed from the estimated advantage:

$$L_V(\phi) = \mathbb{E}_t \big[ (V_\phi(s_t) - \text{stop\_gradient}((V_\phi(s_t) + \hat{A}_t))^2)^2 \big].$$

# 4. Observation: State Value Degeneration

We propose the State Value Estimation Benchmark (SVEB) to evaluate the quality of predicted state values. Using SVEB, we uncover a consistent discrepancy for PPO in the LLM setting: the critic network's value estimates regularly degenerate to a simplistic baseline and fail to provide any informative guidance.

## 4.1. State Value Estimation Benchmark

SVEB evaluates state estimation approaches by computing the error between estimated values and ground-truth values obtained via large-scale Monte Carlo sampling (MCS).

Specifically, the benchmark consists of tuples $\left( s_t, \widehat{V}(s_t) \right)$ where $s_t$ is a state sampled from diverse reasoning domains (in Section 6.1) with a policy LLM and $\widehat{V}(s_t)$ is its reference value estimated by Monte Carlo sampling. The construction procedure for SVEB is detailed below, and key notations with their definitions are summarized in Table 17.

**State collection.** We begin by collecting a dataset of states $\mathcal{D}_s = \{s_t^{(j)}\}_{j=1}^N$. through generating rollouts using a fixed LLM $\pi$ over a set of prompts. For each rollout $\tau = (s_0, a_0, s_1, \ldots, s_T)$,, where $s_0$ is the prompt state and $s_T$ is terminal, we uniformly sample several intermediate states $s_t$ into the dataset $D_s$.

**Monte Carlo reference value.** Since the true state value $V^\pi(s_t)$ is intractable, one way of estimating it is via Monte Carlo sampling. From each $s_t$ in $D_s$, we sample $n$ independent continuations following the same policy $\pi$, yielding terminal rewards $\{r(s_T^{(i)})\}_{i=1}^n$. Let

$$\widehat{V}(s_t) = \frac{1}{n} \sum_{i=1}^n r\left( s_T^{(i)} \right).$$

be the estimated value of $s_t$. By the law of large numbers, $\widehat{V}(s_t) \to V^\pi(s_t)$ almost surely as $n \to \infty$, providing unbiased estimation of true state values.

**Evaluation metric.** Each evaluated state value estimation method is a function $f$ parametrized by $\theta$ that maps the state $s_t$ to its corresponding state value: $\widehat{V}_{f,\theta}(s_t) = f(s_t, \theta)$,. The estimation quality is measured by taking the Mean Absolute Error (MAE) against the MCS reference over $\mathcal{D}_s$:

$$\text{MAE}(f, D_s) = \frac{1}{|D_s|} \sum_{j=1}^{|D_s|} \left| f\left( s_t^{(j)}, \theta \right) - \widehat{V}\left( s_t^{(j)} \right) \right|.$$

Lower MAE indicates better value estimation. The final SVEB score is the average MAE across all sampled states.

## 4.2. PPO on SVEB

We begin by evaluating PPO, a classic actor-critic method that employs a separate critic network for state-value estimation. To isolate the critic's behavior, we freeze the actor and train the critic solely on pre-generated rollouts. In standard PPO, the critic makes predictions without seeing the data during the first epoch, but has seen data after the first epoch. We simulate these two regimes:

**PPO-1:** The critic is evaluated on unseen benchmark data, simulating its behavior of **initial epoch**.

**PPO-N:** The critic is trained and evaluated on the same data, reflecting its performance **after the first epoch**.

We train the PPO's critic network with GAE coefficients $\lambda = 0.95$ and $1.0$ as in (Guo et al., 2025) following state

*Table 1.* SVEB-NUMBER results for PPO-1 and PPO-N. "@n" denotes the number of rollouts used for estimation; reference values are computed via MCS@20.

| METHOD | $\lambda = 0.95$ | $\lambda = 1.0$ | No $\lambda$ |
|---|---|---|---|
| GRPO@40 | - | - | 0.164 |
| PPO-1@40 | 0.169 | 0.161 | - |
| PPO-N@40 | 0.158 | 0.152 | - |

*Table 2.* Performance of GRPO, PPO-N, and *Numca* on different fields of SVEB. The reference value is calculated by MCS@20.

| METHOD | SVEB-NUMBER | SVEB-SCIENCE |
|---|---|---|
| GRPO@40 | 0.164 | 0.215 |
| PPO-1@40 | 0.161 | 0.220 |
| PPO-N@40 | 0.158 | **0.198** |
| NUMCA@40 | **0.136** | 0.217 |

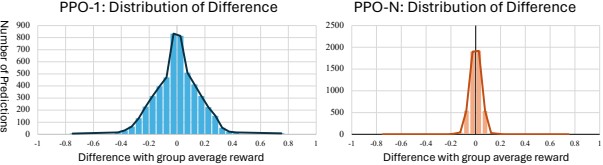

*Figure 2.* Distribution of the difference between state value predicted by PPO with $\lambda = 1.0$ and the group averaged reward.

value definition and training objectives in Section 3.2. Both the actor and critic are initialized from Qwen2.5-1.5B. We construct the SVEB-NUMBER dataset by sampling roughly 1,000 math problems and 5,000 intermediate states from DAPO-17K (Yu et al., 2025) (more details in Section 6.1). The SVEB results (Table 1) show that, in the PPO-1 setting, the critic offers no measurable advantage over the simple group-average baseline used in GRPO. Only after the first epoch (PPO-N) does PPO slightly outperform GRPO on SVEB. Consistent with findings in DeepSeek-R1(Guo et al., 2025), where PPO trained with $\lambda = 1.0$ consistently outperforms the $\lambda = 0.95$ variant in downstream training.

To further investigate the advantage of using a critic network, we plot the difference between the state values estimated by PPO and the group-average baseline used by GRPO, i.e., $\widetilde{V}_{PPO}(s_t) - \widehat{V}_{GRPO}(s_t)$ in Figure 2. The distribution shows that the two estimates are nearly identical, with differences tightly clustered around zero. Although PPO-1 exhibits a larger spread of differences relative to the GRPO baseline, this primarily reflects estimation error from an undertrained critic network. This phenomenon, consistently observed across multiple datasets (see Appendix C for details), reveals a fundamental weakness of the standard TD-based critic in PPO: its value estimates degenerate to the group baseline and fail to provide any fine-grained guidance for RL. Motivated by this limitation, we introduce two methods, *Numca* and *Hista*, to estimate state values more effectively.

## 5. Method

### 5.1. *Numca*

Given that standard critic training fails to provide informative guidance for RL, we instead reframe LLM reasoning as a sparse-reward, goal-reaching task. In this setting, a natural approach inspired by Hindsight Experience Replay (Andrychowicz et al., 2018) is to define intermediate milestones and treat them as verifiable subgoals and states for

value computation. However, applying this idea to language generation is challenging because semantic meaning is distributed across tokens rather than structured into discrete subgoals. We then observe that in mathematical reasoning, numerical values—such as integers, decimals, and fractions—serve as natural, verifiable subgoals. Building on this insight, we propose **Nu**merical **M**ilestone **C**redit **A**ssignment (*Numca*), a method that leverages numerical patterns as structured milestones to guide credit assignment in mathematical problem-solving.

Let $\mathcal{P}$ be a predefined set of numerical patterns (e.g., integers, decimals, fractions). A milestone $m = (x_i, x_{i+1}, ..., x_j)$ is a subsequence of tokens whose decoded string matches a pattern $p \in \mathcal{P}$. Given a state $s_t = (x_1, ..., x_t)$, let $\mathbb{M}(s_t) = \{m_1, ..., m_k\}$ denote all *milestones* that appear in $s_t$. Define the *abstract state* $s_t^M \triangleq \mathbb{M}(s_t)$ that captures only the milestones achieved up to time $t$; A macro action $a^M$ is the sequence of tokens generated between two consecutive abstract states. Formally, if $s_t^M$ and $s_{t'}^M$ $(t' > t)$ are consecutive abstracted states, then the macro action from $s_t^M$ to $s_{t'}^M$ is defined as:

$$a_t^M \triangleq (a_t, a_{t+1}, ..., a_{t'-1}) = (x_{t+1}, x_{t+2}, ..., x_{t'-1})$$

This construction yields a temporally abstracted decision process where each macro action corresponds to solving a subproblem that reaches the next numerical milestone.

*Numca* estimates the value of an abstract state $s^M$ by **averaging the final rewards of all rollouts that contain $s^M$**. This estimated value is then assigned uniformly to every token within the corresponding macro action. The complete procedure is detailed in Algorithm 1 and is implemented using a dictionary, incurring negligible memory and computation overhead. Additional notation explanations are provided in Table 17.

We evaluate *Numca* on SVEB, with results shown in the SVEB-NUMBER column of Table 2, where *Numca* surpasses both PPO-N and GRPO in state value estimation. We further apply *Numca* in RL training using an experimental dataset sampled from DAPO-17K (Yu et al., 2025) and OpenR1-220K (Lozhkov et al., 2025) (see Appendix G.1). The base models are Qwen2.5-1.5B-Instruct and Qwen2.5-1.5B-Math-Instruct (Yang et al., 2024). Training curves are presented in Figure 3, where *Numca* exhibits stable training and consistently improving validation accu-

**Algorithm 1** Numerical Milestone Credit Assignment

**Require:** Rollouts $\mathcal{D}_r$, numerical patterns $\mathcal{P}$
1: Initialize dictionary $\mathcal{T} \leftarrow \emptyset$
   $\{\mathcal{T}[s^M] = (\text{count}, \text{reward\_sum})\}$
2: **for** each rollout $\tau = (s_0, a_0, \ldots, s_T)$ in $\mathcal{D}_r$ **do**
3:   $s^M \leftarrow \emptyset$
4:   **for** $t = 0$ to $T$ **do**
5:     **if** $\mathbb{M}(s_t) \neq s^M$ **then**
6:       $s^M \leftarrow \mathbb{M}(s_t)$
7:       Update $\mathcal{T}[s^M] \leftarrow \mathcal{T}[s^M] + (1, r(s_T))$
8:     **end if**
9:   **end for**
10: **end for**
11: **for** each state abstraction $s^M$ in $\mathcal{T}$ **do**
12:   Estimate state value:

$$V(s^M) \leftarrow \mathcal{T}[s^M].\text{reward\_sum} \, / \, \mathcal{T}[s^M].\text{count}$$

13: **end for**

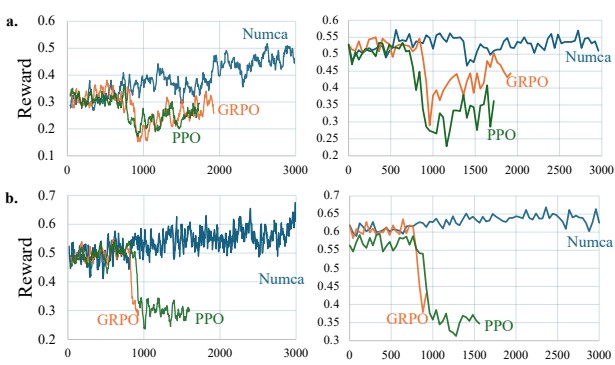

*Figure 3.* Compare reward on train set (left) and validation set (right) of PPO, GRPO, and GRPO + *Numca* during training: a. Qwen2.5-1.5B-Instruct; b. Qwen2.5-1.5B-Math-Instruct.

racy, whereas GRPO and PPO suffer from instability. This is primarily because finer-grained and more accurate credit assignment helps reduce the variance in the gradient. Finally, we select the checkpoint with the best validation accuracy and evaluate it on popular math benchmarks in Table 3. *Numca* can consistently outperform GRPO, demonstrating that efficient state value estimation can effectively benefit RL training.

### 5.2. *Hista*

Despite its strong performance on mathematical reasoning tasks, Numca is difficult to generalize because it relies on numerical patterns for milestone identification. As shown in the SVEB-SCIENCE column of Table 2, and in other multiple-choice scientific QA tasks from Section 6.1 where numerical milestones are rarely present, Numca's performance degrades to that of the GRPO baseline. This indi-

*Table 3.* Performance of Qwen2.5-Math-1.5B-Instuct model, GRPO, and GRPO+*Numca* on math benchmark.

| BENCHMARK | QWEN | +GRPO | +NUMCA |
|---|---|---|---|
| MATH-500 | 0.74 | 0.746 | **0.760** |
| GSM8K | 0.849 | 0.848 | **0.864** |
| MINERVAMATH | 0.286 | 0.301 | **0.313** |
| OLYMPIADBENCH | 0.425 | 0.413 | **0.426** |
| AMC23 | 0.528 | 0.553 | **0.584** |
| AIME24&25 | 0.104 | **0.113** | 0.104 |
| AVERAGE | 0.528 | 0.541 | **0.555** |

cates that Numca is not a general solution for state value estimation in LLM reinforcement learning, as it depends on external knowledge to identify milestone patterns.

This limitation leads us to a broader insight: effective state value estimation requires a state representation capable of **grouping semantically similar but disjoint rollouts** without relying on domain-specific heuristics. Drawing inspiration from representation learning methods such as MAE (He et al., 2021), DINO (Caron et al., 2021), BERT (Devlin et al., 2019), and JEPA (Assran et al., 2023), which learn meaningful embeddings by predicting masked content from context, we posit that auto-regressive language model training is itself a powerful form of representation learning. Consequently, the hidden states of an LLM should serve as a general-purpose state representation. To operationalize this, we measure the distance between two sequence representations using the following MinDistance operation:

**Definition 5.1.** The *MinDistance* (MD) operation between two hidden states $\mathbf{X}_1 = \left[\mathbf{x}_{1,1}, \mathbf{x}_{1,2}, \ldots, \mathbf{x}_{1,\eta_1}\right]^\top$ and $\mathbf{X}_2 = \left[\mathbf{x}_{2,1}, \mathbf{x}_{2,2}, \ldots, \mathbf{x}_{2,\eta_2}\right]^\top$ is:

$$\begin{cases} \sum_{i=1}^{\eta_1} (\min_{j=1,\ldots,\eta_2} \|\mathbf{x}_{1,i} - \mathbf{x}_{2,j}\|_2), & \eta_1 \geq \eta_2, \\ \sum_{i=1}^{\eta_2} (\min_{j=1,\ldots,\eta_1} \|\mathbf{x}_{1,j} - \mathbf{x}_{2,i}\|_2), & \eta_1 < \eta_2. \end{cases}$$

We establish a theoretical correlation between MD and the likelihood of two states sharing the same final reward.

**Theorem 5.2** (Correlation between Hidden States and Reward). *Suppose $R_i$ is the final reward of generated sequence from state $s_i$, which fulfill the Assumption A.1. Then, the probability that two states receive the same reward $P(R_1 = R_2)$ is correlated with the MinDistance between their hidden state representations:*

$$P(R_1 = R_2) \propto \frac{1}{\text{MD}(\mathbf{X}_1^l, \mathbf{X}_2^l)}, \quad \forall l \in [1, L]$$

The proof is in Appendix A. With this correlation, we can prove that weighted estimator based on hidden states will produce better state value than GRPO. Consider a set of state and final reward pair $\{(s_0, r_0), \ldots, (s_\mathcal{N}, r_\mathcal{N})\}$ We compare two methods for estimating the value of a state $s_t$:

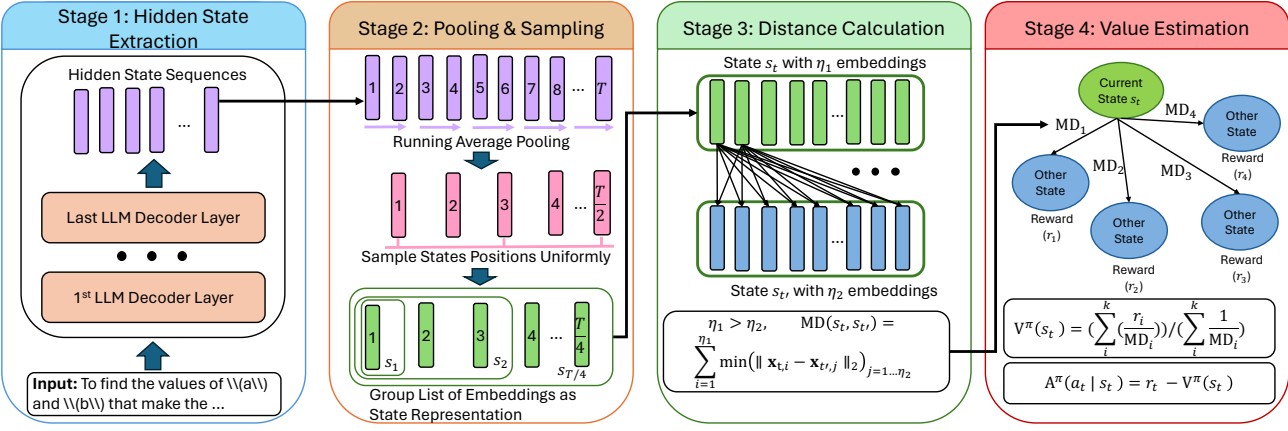

*Figure 4.* Pipeline of *Hista*. It takes the last layer hidden state and pools them to state representation. Then, *MinDist* operation is applied to calculate distance between two states, which is weighted averaged to get the state value estimation. Finally, the estimated state value is treated as baseline to calculate advantage.

**Definition 5.3.** The average estimator: $\hat{V}_{\text{avg}}(s_t) = \frac{1}{\mathcal{N}} \sum_{i=1}^{\mathcal{N}} r_i$, which is an approximation of group average reward.

**Definition 5.4.** The probability-weighted estimator: $\hat{V}_{\text{PW}}(s_t) = \frac{\sum_{i=1}^{\mathcal{N}} P_{t,i} r_i}{\sum_{i=1}^{\mathcal{N}} P_{t,i}}$, where $P_{t,i} = P(R_t = R_i)$ is obtained from Theorem 5.2.

**Theorem 5.5** (Superiority of Probability-Weighted Estimation). *Assume probability weights satisfy the Assumption B.2. Then probability-weighted estimator has no larger absolute bias than the naive average estimator:*

$$\left| \mathbb{E}[\hat{V}_{\text{PW}}(s_t)] - V(s_t) \right| \leq \left| \mathbb{E}[\hat{V}_{\text{avg}}(s_t)] - V(s_t) \right|.$$

The detailed proof of Theorem 5.5 and remark about Assumption B.2 is provided in Remark B.4.

Building on this insight, we propose Hidden State based State Value Estimation (*Hista*), which implements probability weighted estimation to compute state values from generation rollouts. For efficiency, *Hista* extracts last layer hidden states and uses the inverse of their *MinDistance* to approximate the probability score. As shown in Figure 4, *Hista* directly process the terminal state $s_T$ of rollout $\tau$ and extract the representation of other states from it. Suppose the last-layer hidden states of $s_T$ is $\mathbf{X}_\tau = [\mathbf{x}_{\tau,1}, \mathbf{x}_{\tau,2}, ..., \mathbf{x}_{\tau,\eta}]^\top$. Since this sequence can be very long (e.g., $> 100$k tokens), we compress it by applying exponential moving average (EMA) with smoothing factor $\alpha$ and a compression interval $\varphi$ on the first dimension. This yields a shorter embedding sequence $\mathbf{E}_\tau = [\mathbf{e}_{\tau,1}, \mathbf{e}_{\tau,2}, \ldots, \mathbf{e}_{\tau,\lfloor \eta/\varphi \rfloor}]^\top$, where $\mathbf{e}_{\tau,i}$ is the last embedding of $\text{EMA}([\mathbf{x}_{\tau,1}, \mathbf{x}_{\tau,2}, ..., \mathbf{x}_{\tau,i\varphi}]^\top)$. We then uniformly sample embedding positions with interval $\delta$ to define the finite state space for this rollout $\mathcal{S}_\tau^H = (s_{\tau,1}^H, s_{\tau,2}^H, \ldots, s_{\tau,\lfloor \eta/(\varphi\delta) \rfloor}^H)$. Each state $s_{\tau,i}^H$ is rep-

resented by the sequence of embeddings up to that point $s_{\tau,i}^H = [\mathbf{e}_{\tau,1}, \mathbf{e}_{\tau,2}, \ldots, \mathbf{e}_{\tau,i\delta}]^\top$.

All states sharing the same input prompt are collected into a global state space $\mathcal{S}^H$. For a given state $s_t^H$, we compute its distance to every other state in $\mathcal{S}^H$ using the *MinDistance* measure, select the $k$ nearest neighbors, and record their distances. Finally, as illustrated in Stage 4 in Figure 4, the state value is estimated as the probability-weighted average of the final rewards of these neighbors:

$$V(s_t) = \frac{\sum_{i=1}^{k} \omega_i \cdot r_i}{\sum_{i=1}^{k} \omega_i}, \quad \omega_i = \frac{1}{\text{MD}(s_t, s_i)}.$$

Hyperparameter ablations are provided in Appendix E. With batched GPU computation, *Hista* achieves efficiency comparable to GRPO, with space complexity $O(\lfloor T/d \rfloor^2)$, corresponding latency and memory results are reported in the Appendix F. Notation definitions appear in Table 17.

## 6. Experiments

### 6.1. Experiment Settings

The SVEB is created by curating data from 5 specialized sources that coverage most of reasoning tasks. To calibrate problem difficulty, we retain only problems with a solution accuracy between 0.1 and 0.8.

**SVEB-Number**: Samples are drawn from DAPO-17K (Yu et al., 2025), filtered to include only problems where the number of unique numerical values appearing exclusively within the reasoning trajectory exceeds four.

**SVEB-Math**: This subset consists of general mathematical problems from OpenR1-220K (Lozhkov et al., 2025).

**SVEB-Science**: Sampled from the science subset of the

*Table 4.* Mean Absolute Error (MAE) for GRPO, PPO-N, *Numca*, and *Hista* across SVEB fields, with lower values being better. All values are calculated relative to a reference standard established by MCS@20.

| SVEB | Number ↓ | Math ↓ | Science ↓ | General ↓ | Programming ↓ |
|---|---|---|---|---|---|
| GRPO@40 | 0.175 | 0.208 | 0.215 | 0.202 | 0.157 |
| PPO-N@40 | 0.159 | 0.187 | 0.198 | 0.185 | 0.144 |
| Numca@40 | **0.132**$_{\downarrow 0.027}$ | 0.194$_{\uparrow 0.007}$ | 0.217$_{\uparrow 0.019}$ | 0.200$_{\uparrow 0.015}$ | 0.154$_{\uparrow 0.010}$ |
| Hista@40 | 0.142$_{\downarrow 0.017}$ | **0.145**$_{\downarrow 0.042}$ | **0.173**$_{\downarrow 0.025}$ | **0.157**$_{\downarrow 0.028}$ | **0.119**$_{\downarrow 0.025}$ |
| MCS@1 | 0.223 | 0.235 | 0.272 | 0.283 | 0.162 |
| MCS@2 | 0.133 | 0.160 | 0.188 | 0.139 | 0.113 |
| MCS@3 | 0.111 | 0.121 | 0.134 | 0.114 | 0.083 |

*Table 5.* DAPO, GSPO, CSIPO, and their combination with *Hista* on training **Qwen2.5 base models** with Simple Math Dataset.

| | MATH | GSM | Minerva | Olympiad | AMC | AIME | GaoKao | CollegeMath | Average |
|---|---|---|---|---|---|---|---|---|---|
| 0.5B DAPO | 28.4 | 45.3 | 5.3 | 7.0 | 7.8 | 0.2 | 28.0 | 34.4 | 19.6 |
| +Hista | 32.5$_{\uparrow 4.1}$ | 47.4$_{\uparrow 2.1}$ | 6.4$_{\uparrow 1.1}$ | 7.8$_{\uparrow 0.9}$ | 9.3$_{\uparrow 1.5}$ | 0.2$_{\downarrow 0.0}$ | 30.6$_{\uparrow 2.6}$ | 35.4$_{\uparrow 1.0}$ | 21.2$_{\uparrow 1.8}$ |
| 1.5B DAPO | 57.6 | 74.2 | 18.8 | 20.2 | 23.1 | 2.0 | 49.4 | 53.3 | 37.3 |
| +Hista | 55.8$_{\downarrow 1.8}$ | 76.4$_{\uparrow 2.2}$ | 19.4$_{\uparrow 0.6}$ | 20.6$_{\uparrow 0.4}$ | 27.2$_{\uparrow 4.1}$ | 2.1$_{\uparrow 0.1}$ | 51.2$_{\uparrow 1.8}$ | 51.5$_{\downarrow 1.8}$ | 38.0$_{\uparrow 0.7}$ |
| 1.5B GSPO | 57.0 | 74.1 | 17.7 | 20.0 | 30.6 | 2.0 | 49.3 | 52.6 | 37.9 |
| +Hista | 55.8$_{\downarrow 1.2}$ | 74.6$_{\uparrow 0.5}$ | 17.9$_{\uparrow 0.2}$ | 22.6$_{\uparrow 2.6}$ | 26.3$_{\downarrow 4.3}$ | 2.3$_{\uparrow 0.3}$ | 49.8$_{\uparrow 0.5}$ | 52.5$_{\downarrow 0.1}$ | 37.7$_{\downarrow 0.2}$ |
| 1.5B CSIPO | 54.0 | 74.1 | 19.2 | 20.6 | 32.8 | 1.2 | 48.3 | 52.4 | 37.8 |
| +Hista | 58.2$_{\uparrow 4.2}$ | 74.2$_{\uparrow 0.1}$ | 18.0$_{\downarrow 1.2}$ | 21.0$_{\uparrow 0.4}$ | 35.3$_{\uparrow 2.5}$ | 1.0$_{\downarrow 0.2}$ | 48.1$_{\downarrow 0.2}$ | 53.5$_{\uparrow 1.1}$ | 38.7$_{\uparrow 0.9}$ |
| 3B DAPO | 66.0 | 83.6 | 25.8 | 29.8 | 37.1 | 3.9 | 58.2 | 57.6 | 45.2 |
| +Hista | 65.6$_{\downarrow 0.4}$ | 86.4$_{\uparrow 2.8}$ | 24.9$_{\downarrow 0.9}$ | 30.1$_{\uparrow 0.3}$ | 39.7$_{\uparrow 2.6}$ | 5.6$_{\uparrow 1.7}$ | 55.6$_{\downarrow 2.6}$ | 58.1$_{\uparrow 0.5}$ | 45.8$_{\uparrow 0.6}$ |
| 3B CSIPO | 64.4 | 83.6 | 28.4 | 28.6 | 35.0 | 3.8 | 52.7 | 59.0 | 44.4 |
| +Hista | 62.0$_{\downarrow 2.4}$ | 84.2$_{\uparrow 0.6}$ | 25.7$_{\downarrow 2.7}$ | 31.8$_{\uparrow 3.2}$ | 35.9$_{\uparrow 0.9}$ | 6.4$_{\uparrow 2.6}$ | 55.8$_{\uparrow 3.1}$ | 57.8$_{\downarrow 1.2}$ | 45.0$_{\uparrow 0.6}$ |
| 7B DAPO | 75.4 | 91.1 | 32.1 | 39.9 | 47.8 | 11.0 | 63.8 | 63.8 | 53.1 |
| +Hista | 72.6$_{\downarrow 2.8}$ | 91.5$_{\uparrow 0.4}$ | 33.1$_{\uparrow 1.0}$ | 41.2$_{\uparrow 1.3}$ | 48.8$_{\uparrow 1.0}$ | 12.0$_{\uparrow 1.0}$ | 65.4$_{\uparrow 1.6}$ | 64.3$_{\uparrow 0.5}$ | 53.6$_{\uparrow 0.5}$ |
| 7B CSIPO | 76.4 | 90.0 | 31.3 | 40.5 | 45.0 | 9.1 | 64.4 | 62.3 | 52.4 |
| +Hista | 74.2$_{\downarrow 2.2}$ | 89.5$_{\downarrow 0.5}$ | 34.2$_{\uparrow 2.9}$ | 38.8$_{\downarrow 1.7}$ | 50.6$_{\uparrow 5.6}$ | 9.4$_{\uparrow 0.3}$ | 63.9$_{\downarrow 0.5}$ | 62.9$_{\uparrow 0.6}$ | 52.9$_{\uparrow 0.5}$ |
| 14B DAPO | 77.6 | 94.2 | 38.1 | 46.1 | 55.6 | 12.9 | 68.5 | 64.7 | 57.2 |
| +Hista | 78.0$_{\uparrow 0.4}$ | 94.4$_{\uparrow 0.2}$ | 38.5$_{\uparrow 0.4}$ | 45.5$_{\downarrow 0.6}$ | 61.8$_{\uparrow 6.2}$ | 12.1$_{\downarrow 0.8}$ | 71.1$_{\uparrow 2.6}$ | 65.3$_{\uparrow 0.6}$ | 58.3$_{\uparrow 1.1}$ |

Llama-Nemotron Post-Training Dataset (Bercovich et al., 2025) to evaluate reasoning in scientific domains.

**SVEB-General**: Samples are drawn from the WebInstruct-verified dataset (Ma et al., 2025) to assess performance on general question-answering tasks.

**SVEB-Programming**: This subset is constructed from the Python portion of the verifiable-coding-problem dataset (Mattern et al., 2025), targeting evaluation of state value estimation in code-oriented reasoning.

**RL Settings**: We evaluate *Hista* by integrating it into RL post-training across diverse datasets. First, we treat MATH dataset (Hendrycks et al., 2021) as our **Simple Math Dataset** to train Qwen2.5 models (1.5B, 3B, and 7B). Second, we construct a **Hybrid Reasoning Dataset** by aggregating and filtering all data sources from Section 6.1 (see Appendix H for details). Finally, for advanced reasoning,

we train R1-Distill-1.5B on the challenging DAPO-17K dataset (**Hard Math Dataset**). Additional training details are provided in Appendix G.2.

**Downstream Task Benchmarks**: To evaluate different algorithms in alignment with our training tasks, we employ comprehensive suite of benchmarks. For mathematical reasoning, we use MATH-500(Lightman et al., 2023), GSM8K (Cobbe et al., 2021), OlympiadBench (He et al., 2024), AIME24&25, AMC23, GaoKao-2023-en (Zhang et al., 2024), and CollegeMath (Tang et al., 2024). In the scientific domain, models are assessed on SciEval (Sun et al., 2024), TheoremQA (Chen et al., 2023), and Minerva-Math (Lewkowycz et al., 2022). For general knowledge and reasoning, we utilize GPQA (Rein et al., 2023), and MMLU-Pro (Wang et al., 2024). Finally, programming capability is evaluated using HumanEval+ (Chen et al., 2021)(Liu et al., 2023) and MBPP+(Austin et al., 2021)(Liu et al., 2023).

*Table 6.* Performance of DAPO, CSIPO, and their combination with *Hista* on training **Qwen2.5 instruct models** with Hybrid Reasoning Dataset. The "... Average" column is calculated by averaging the score of all benchmarks with same category. The "ALL" column is calculated by averaging the performance on all four fields. Each "... + Hista" row is compared with corresponding "..." baseline.

| | Math Average | Science Average | General Average | Programming Average | ALL |
|---|---|---|---|---|---|
| Qwen2.5-1.5B | 39.3 | 34.9 | 26.8 | 53.5 | 38.6 |
| DAPO | 38.4 | 35.8 | 29.0 | 52.7 | 38.9 |
| CSIPO | 40.4 | 36.1 | 26.0 | 53.9 | 39.1 |
| DAPO + Hista | $40.1_{\uparrow 1.7}$ | $36.6_{\uparrow 0.8}$ | $29.5_{\uparrow 0.5}$ | $53.4_{\uparrow 0.7}$ | $39.9_{\uparrow 1.0}$ |
| CSIPO + Hista | $41.5_{\uparrow 1.1}$ | $37.2_{\uparrow 1.1}$ | $30.0_{\uparrow 4.0}$ | $54.6_{\uparrow 0.7}$ | $40.8_{\uparrow 1.7}$ |
| Qwen2.5-3B | 48.3 | 43.1 | 35.9 | 59.7 | 46.8 |
| DAPO | 47.0 | 42.5 | 36.1 | 58.4 | 46.0 |
| CSIPO | 49.5 | 45.3 | 36.8 | 60.1 | 47.9 |
| DAPO + Hista | $49.2_{\uparrow 2.2}$ | $44.0_{\uparrow 1.5}$ | $37.1_{\uparrow 1.0}$ | $59.2_{\uparrow 0.8}$ | $47.4_{\uparrow 1.4}$ |
| CSIPO + Hista | $50.4_{\uparrow 0.9}$ | $44.3_{\downarrow 1.0}$ | $37.1_{\uparrow 0.3}$ | $61.0_{\uparrow 0.9}$ | $48.2_{\uparrow 0.3}$ |
| Qwen2.5-7B | 57.6 | 48.3 | 45.7 | 68.1 | 54.9 |
| DAPO | 56.5 | 50.0 | 44.4 | 70.0 | 55.2 |
| CSIPO | 56.6 | 51.7 | 45.3 | 69.1 | 55.7 |
| DAPO + Hista | $59.0_{\uparrow 2.5}$ | $51.5_{\uparrow 1.5}$ | $46.1_{\uparrow 1.7}$ | $68.6_{\downarrow 1.4}$ | $56.3_{\uparrow 1.1}$ |
| CSIPO + Hista | $57.2_{\uparrow 0.6}$ | $52.0_{\uparrow 0.3}$ | $45.8_{\uparrow 0.5}$ | $68.7_{\downarrow 0.4}$ | $55.9_{\uparrow 0.2}$ |
| Qwen2.5-14B | 61.6 | 56.7 | 53.2 | 68.8 | 60.1 |
| DAPO | 60.1 | 56.7 | 54.6 | 69.8 | 60.3 |
| +Hista | $61.2_{\uparrow 1.1}$ | $57.4_{\uparrow 0.7}$ | $55.6_{\uparrow 1.0}$ | $69.8_{\uparrow 0.0}$ | $61.0_{\uparrow 0.7}$ |

*Table 7.* Performance of DAPO and its combination with *Hista* on training **reasoning model** with Hard Math Dataset.

| | MATH | GSM | Minerva | Olympiad | AMC | AIME | GaoKao | CollegeMath | Average |
|---|---|---|---|---|---|---|---|---|---|
| R1-Dist-1.5B | 82.6 | 76.4 | 29.7 | 52.2 | 72.5 | 25.0 | 70.8 | 62.1 | 58.9 |
| DAPO | 86.0 | 79.3 | 31.6 | 53.2 | 71.8 | 25.4 | 72.5 | 62.5 | 60.3 |
| +Hista | $85.0_{\downarrow 1.0}$ | $79.7_{\uparrow 0.4}$ | $32.3_{\uparrow 0.7}$ | $53.5_{\uparrow 0.3}$ | $76.8_{\uparrow 5.0}$ | $26.3_{\uparrow 0.9}$ | $73.4_{\uparrow 0.9}$ | $63.8_{\uparrow 1.3}$ | $61.4_{\uparrow 1.1}$ |

To ensure efficiency, stability, and comparability with the original model, most benchmarks are evaluated with temperature 0.0. For smaller benchmark sets such as AIME and AMC, we use temperature = 0.1 and report avg@16. Different from benchmark evaluation, the validation dynamics in Figure 1 and 5 are obtained with temperature = 0.7, which might be the reason for some difference between validation result and MATH column of Table 5.

## 6.2. Results

**SVEB Results**: As shown in Table 4, *Hista* achieves dominant and generalized performance when applied to the Qwen2.5-3B-Instruct actor, attaining a lower MAE across most SVEB benchmarks. While *Numca* excels narrowly on SVEB-NUMBER, it fails to generalize to other reasoning tasks. These results validate our claim that both *Numca* and *Hista* provide significantly more accurate state value estimates than mainstream alternatives such as GRPO and PPO. Additional results on different model sizes and architectures are provided in the Appendix D. Compared to Monte Carlo Search (MCS) estimation, GRPO outperforms MCS@1,

while *Hista* matches the accuracy of the more expensive MCS@2. However, no current method reaches the performance ceiling set by MCS@3, leaving substantial room for future improvement. The distribution of differences relative to GRPO is presented in Appendix C.

**RL Training with *Hista* on Base Model**: We evaluate *Hista* on base models ranging from 0.5B to 14B parameters. As shown in Table 5, integrating *Hista* with DAPO yields consistent and stable accuracy gains across all model sizes, demonstrating strong synergy between *Hista*'s token-level MDP modeling and DAPO's design. In contrast, combining *Hista* with GSPO on a 1.5B model offers no improvement due to a fundamental conflict with GSPO's whole-response optimization. Meanwhile, *Hista* consistently outperforms CSIPO, highlighting the advantage of its structurally aligned MDP formulation.

**RL Training with *Hista* on Instruct Model**: We extend our evaluation to instruction-tuned models trained on hybrid reasoning datasets. As presented in Table 6, applying RL with *Hista* to these models yields further improvements compare with original model. This finding aligns with the results in

*Table 8.* DAPO and its combination with *Hista* on training **Qwen3 base models**. 0.6B model is trained on Simple Math Dataset. 1.7B, 4B, and 8B models are trained on Hard Math dataset

|  | MATH | GSM | Minerva | Olympiad | AMC | AIME | GaoKao | CollegeMath | Average |
|---|---|---|---|---|---|---|---|---|---|
| 0.6B DAPO | 50.8 | 67.2 | 17.0 | 17.1 | 18.1 | 1.8 | 41.6 | 46.9 | 32.5 |
| +Hista | $52.2_{\uparrow 1.4}$ | $68.3_{\uparrow 1.1}$ | $17.6_{\uparrow 0.6}$ | $19.1_{\uparrow 2.0}$ | $26.1_{\uparrow 8.0}$ | $1.7_{\downarrow 0.1}$ | $43.6_{\uparrow 2.0}$ | $49.3_{\uparrow 2.4}$ | $34.8_{\uparrow 2.3}$ |
| 1.7B DAPO | 65.6 | 83.8 | 26.6 | 30.8 | 41.7 | 5.0 | 57.7 | 59.2 | 46.3 |
| +Hista | $67.2_{\uparrow 1.6}$ | $81.4_{\downarrow 2.4}$ | $22.9_{\downarrow 3.7}$ | $30.3_{\downarrow 0.5}$ | $41.0_{\downarrow 0.7}$ | $7.5_{\uparrow 2.5}$ | $58.4_{\uparrow 0.7}$ | $58.9_{\downarrow 0.3}$ | $45.9_{\downarrow 0.4}$ |
| 4B DAPO | 79.8 | 88.6 | 36.4 | 45.5 | 52.2 | 15.4 | 67.2 | 63.3 | 56.1 |
| +Hista | $80.2_{\uparrow 0.4}$ | $90.2_{\uparrow 1.6}$ | $33.0_{\downarrow 3.4}$ | $45.7_{\uparrow 0.2}$ | $56.3_{\uparrow 4.1}$ | $17.8_{\uparrow 2.4}$ | $68.3_{\uparrow 1.1}$ | $64.4_{\uparrow 1.1}$ | $57.4_{\uparrow 1.3}$ |
| 8B DAPO | 83.0 | 93.3 | 40.2 | 51.8 | 58.8 | 20.2 | 74.0 | 64.8 | 60.7 |
| +Hista | $86.0_{\uparrow 3.0}$ | $93.3_{\uparrow 0.0}$ | $41.3_{\uparrow 1.1}$ | $55.0_{\uparrow 3.2}$ | $58.4_{\downarrow 0.4}$ | $22.5_{\uparrow 2.3}$ | $74.3_{\uparrow 0.3}$ | $66.4_{\uparrow 1.6}$ | $62.2_{\uparrow 1.5}$ |

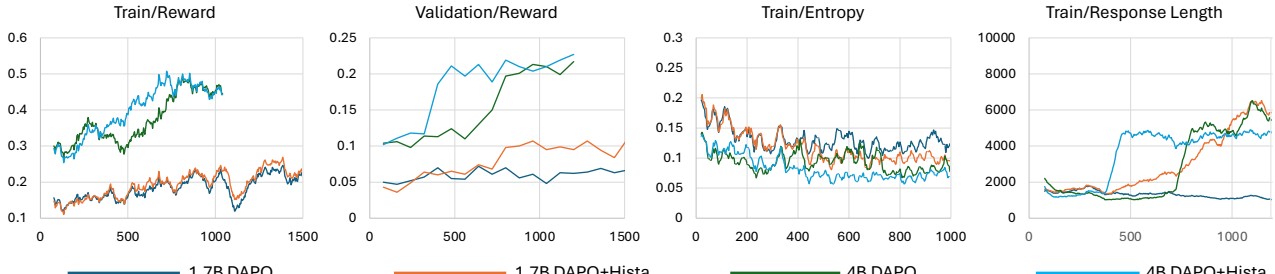

*Figure 5.* Training dynamics of **Qwen3-1.7B and 4B base model** on DAPO-17K dataset and AIME24 validation set.

Table 3, which independently shows that instruction-tuned models obtain greater performance gains from subsequent RL training. The robustness of *Hista* is further confirmed by its reliable enhancement of the DAPO and CSIPO baseline across diverse domains including science, general QA, and programming within this instructed setting. This consistent cross-domain improvement is corroborated by the state-value estimation metrics in Table 4.

**Reasoning Model and Qwen3 Training**: We evaluate *Hista* on long-sequence reasoning and complex mathematical induction (Table 7), where it improves upon the DAPO baseline, empirically validating Theorem 5.5 in reasoning scenario. Applied to Qwen3 models (0.6B–8B) on Simple/Hard Math datasets, *Hista* stimulates reasoning more efficiently and yields more stable entropy (Figure 5). Benchmark results (Table 8) further confirm its generality, though it underperforms on the 1.7B mode. We attribute this discrepancy to the stochasticity inherent in RL and the out-of-distribution (O.O.D) nature of the benchmark.

## 7. Discussion

**Agentic RL:** Agentic RL has became an emerging topic because of its application on programming, research, and task automation. Compared with single turn RL, Agentic RL introduce higher rollout cost because of longer reasoning trace and multiple interaction with tools. Better credit assignment has been proved to be effective in Agentic RL (Feng et al., 2025), but current methods still rely on heuris-

tic rules. Therefore, *Hista* can be seamlessly integrated Agentic RL for provably better credit assignment without extra rollout or combined with other methods as a weak interaction-level supervision (Zou et al., 2026b;a).

**Promgramming Task:** Theorem 5.2 required Assumption A.1, which is not applicable to open-ended programming task. However, the SVEB result in Table 4 prove the effective of *Hista* on programming, which suggests the generality of *Hista* in open-ended reasoning task and the potential of developing a more general theoritical framework.

## 8. Conclusion

This study demonstrates that advancing state-value estimation beyond group-averaged rewards is important for improving reinforcement learning in large language models. Our SVEB framework exposes this limitation when applying conventional critic training approach to LLMs. We then address the issue with two credit assignment approaches: the heuristic *Numca* and the theoretically grounded, token-level *Hista*. Extensive experiments confirm that *Hista* effectively enhances policy optimization. These findings establish superior state-value estimation as a promising direction for boosting the efficiency and performance of RL training.

## Impact Statement

This paper presents work whose goal is to advance the field of Machine Learning. There are many potential societal

consequences of our work, none which we feel must be specifically highlighted here.

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

# A. Proof of Theorem 5.2

## A.1. Preliminaries

We conduct our theoretical analysis within the framework of the standard pre-normalization (Pre-LN) Transformer block, which serves as the architectural foundation for modern open-source large language models such as LLaMA (Touvron et al., 2023) and Qwen. Formally, let the hidden representations at layer $l - 1$ be denoted by the matrix

$$\mathbf{X}^{l-1} = \left[\mathbf{x}_1^{l-1}, \mathbf{x}_2^{l-1}, \ldots, \mathbf{x}_\eta^{l-1}\right]^\top \in \mathbb{R}^{\eta \times d},$$

where $\eta$ represents the sequence length and $d$ signifies the hidden dimension.

**Self-Attention Sublayer.** The hidden state matrix is initially normalized via Root Mean Square Layer Normalization (RMSNorm):

$$\widetilde{\mathbf{x}}_t^{l-1} = \mathrm{RMSNorm}(\mathbf{x}_t^{l-1}) = \mathbf{G}^l \frac{\mathbf{x}_t^{l-1}}{\sqrt{\frac{1}{d}\|\mathbf{x}_t^{l-1}\|_2^2 + \epsilon}},$$

where $\mathbf{G}^l = \mathrm{diag}(\mathbf{g}^l) \in \mathbb{R}^{d \times d}$ represents the diagonal matrix formed by the learnable scaling vector $\mathbf{g}^l \in \mathbb{R}^d$. The corresponding query, key, and value matrices are subsequently obtained through linear projections on $\widetilde{\mathbf{X}}^{l-1} = \left[\widetilde{\mathbf{x}}_1^{l-1}, \widetilde{\mathbf{x}}_2^{l-1}, \ldots, \widetilde{\mathbf{x}}_\eta^{l-1}\right]^\top$:

$$\mathbf{Q}^l = \widetilde{\mathbf{X}}^{l-1}\mathbf{W}_Q^l, \quad \mathbf{K}^l = \widetilde{\mathbf{X}}^{l-1}\mathbf{W}_K^l, \quad \mathbf{V}^l = \widetilde{\mathbf{X}}^{l-1}\mathbf{W}_V^l,$$

where $\mathbf{W}_Q^l, \mathbf{W}_K^l, \mathbf{W}_V^l \in \mathbb{R}^{d \times d}$ denote the learnable projection weight matrices.

To streamline our theoretical exposition without loss of generality, positional encodings, attention masks, and multi-head mechanisms are omitted. Under this single-head self-attention setting, the attention weight matrix $\mathbf{A}^l \in \mathbb{R}^{\eta \times \eta}$ is formulated as

$$\mathbf{A}^l = \mathrm{Softmax}\left(\frac{\mathbf{Q}^l(\mathbf{K}^l)^\top}{\sqrt{d}}\right),$$

where $\mathrm{Softmax}(\cdot)$ denotes the row-wise softmax operator. The attention output matrix before and after the linear output projection $\mathbf{W}_O^l \in \mathbb{R}^{d \times d}$ is given by:

$$\mathbf{H}^l = \mathbf{A}^l\mathbf{V}^l, \quad \mathbf{Z}^l = \mathbf{H}^l\mathbf{W}_O^l.$$

Finally, a residual connection integrates the sublayer output to produce the intermediate representation matrix $\mathbf{U}^l \in \mathbb{R}^{\eta \times d}$:

$$\mathbf{U}^l = \mathbf{X}^{l-1} + \mathbf{Z}^l.$$

**Feed-Forward Sublayer.** The intermediate representation matrix is normalized using the secondary RMSNorm layer:

$$\widetilde{\mathbf{U}}^l = \mathrm{RMSNorm}_2(\mathbf{U}^l).$$

The feed-forward network (FFN) is applied position-wise across the sequence. Maintaining a uniform domain $d \times d$ for structural simplicity, the sublayer operation is expressed as

$$\mathrm{FFN}(\widetilde{\mathbf{U}}^l) = \sigma\left(\widetilde{\mathbf{U}}^l\mathbf{W}_1^l + \mathbf{b}_1^l\right)\mathbf{W}_2^l + \mathbf{b}_2^l,$$

where $\mathbf{W}_1^l, \mathbf{W}_2^l \in \mathbb{R}^{d \times d}$ denote the projection weights, $\mathbf{b}_1^l, \mathbf{b}_2^l$ represent the bias vectors (added row-wise via standard broadcasting), and $\sigma(\cdot)$ denotes an element-wise non-linear activation function.

The final output matrix of layer $l$, denoted as $\mathbf{X}^l \in \mathbb{R}^{\eta \times d}$, is obtained via the second residual connection:

$$\mathbf{X}^l = \mathbf{U}^l + \mathrm{FFN}(\widetilde{\mathbf{U}}^l).$$

## A.2. Correlation in the Simplified Case

Our primary objective is to establish the correlation between the *MinDistance* of the hidden states of two distinct states, $s_1$ and $s_2$, and the probability that they share an identical final reward. Let $\mathbf{X}_1^l, \mathbf{X}_2^l \in \mathbb{R}^{\eta \times d}$ denote the hidden state sequences of $s_1$ and $s_2$ at the $l$-th layer, respectively. For tractability, we initially consider a simplified setting where both sequences share a uniform length $\eta$, and their spatial alignment satisfies:

$$i = \arg\min_{j \in \{1, \ldots, \eta\}} \|\mathbf{x}_{1,i}^l - \mathbf{x}_{2,j}^l\|_2, \quad \forall i \in \{1, \ldots, \eta\}. \quad (1)$$

This alignment condition implies that for each token position $i$ in $\mathbf{X}_1^l$, the closest hidden state in $\mathbf{X}_2^l$ resides at the exact same positional index. Consequently, under this structural assumption, the minimum distance metric reduces to the entry-wise alignment sum:

$$\mathrm{MD}(\mathbf{X}_1^l, \mathbf{X}_2^l) = \sum_{i=1}^\eta \|\mathbf{x}_{1,i}^l - \mathbf{x}_{2,i}^l\|_2. \quad (2)$$

To analyze the trajectory of these representations within this simplified setting, we introduce two foundational assumptions regarding the generation dynamics and latent boundaries beyond the prefix sequence.

**Assumption A.1** (Answer Parsing). The final answer for both states is parsed from one action $a$, which corresponds to one token. This structural setup typically aligns with standard evaluations in Math, Science, and General QA domains, though it generally does not apply to open-ended programming tasks.

**Assumption A.2** (Answer Index). The final answer for both states is parsed from the action located at token index $\ell$.

The proof proceeds inductively by demonstrating that after passing through each constitutive component of a single Transformer block, the divergence between the $a_{T-1}$ of both states—the representation that ultimately dictates the final reward—remains bounded under our assumptions. This layer-wise bound can then be recursively applied across the network architecture. Since Lemmas A.5 through A.9 hold generally for any arbitrary layer, we temporarily omit the layer superscript $l$ for the remainder of this section to streamline notation. We begin our analysis with the first component, RMSNorm:

**Lemma A.3** (Lipschitz Continuity of RMSNorm). *For any* $\mathbf{x}_1, \mathbf{x}_2 \in \mathbb{R}^d$, *the Root Mean Square Layer Normalization satisfies the following Lipschitz bound:*

$$\|\text{RMSNorm}(\mathbf{x}_1) - \text{RMSNorm}(\mathbf{x}_2)\|_2 \leq \frac{\|\mathbf{g}\|_\infty}{\sqrt{\epsilon}} \|\mathbf{x}_1 - \mathbf{x}_2\|_2,$$

*where* $\|\mathbf{g}\|_\infty = \max_i |g_i^l|$ *is the maximum absolute value of the scaling weights.*

*Proof.* Define the unscaled normalized mapping as $y(\mathbf{x}) = \frac{\mathbf{x}}{r(\mathbf{x})}$, where $r(\mathbf{x}) = \sqrt{\frac{1}{d}\|\mathbf{x}\|_2^2 + \epsilon}$. By definition of RMSNorm, we have

$$\begin{aligned}
&\|\text{RMSNorm}(\mathbf{x}_1) - \text{RMSNorm}(\mathbf{x}_2)\|_2 \\
&= \|\mathbf{G}(y(\mathbf{x}_1) - y(\mathbf{x}_2))\|_2 \\
&\leq \|\mathbf{g}\|_\infty \|y(\mathbf{x}_1) - y(\mathbf{x}_2)\|_2.
\end{aligned}$$

To establish a tight Lipschitz constant for $y(\mathbf{x})$, we evaluate the operator norm of its Jacobian matrix $J_y(\mathbf{x}) \in \mathbb{R}^{d \times d}$. Differentiating $y(\mathbf{x})$ with respect to $\mathbf{x}$ using the quotient rule yields:

$$J_y(\mathbf{x}) = \frac{1}{r(\mathbf{x})}\mathbf{I}_d - \frac{\mathbf{x}}{r(\mathbf{x})^2}(\nabla r(\mathbf{x}))^T.$$

The gradient of the root mean square operator is $\nabla r(\mathbf{x}) = \frac{\mathbf{x}}{d\,r(\mathbf{x})}$. Substituting this back into the expression gives:

$$J_y(\mathbf{x}) = \frac{1}{r(\mathbf{x})}\left(\mathbf{I}_d - \frac{\mathbf{x}\mathbf{x}^T}{d\,r(\mathbf{x})^2}\right).$$

We now determine the spectral norm (maximum singular value) of $J_y(\mathbf{x})$, denoted as $\|J_y(\mathbf{x})\|_2$. Let $\mathbf{u} \in \mathbb{R}^d$ be an arbitrary vector such that $\|\mathbf{u}\|_2 = 1$. We decompose $\mathbf{u}$ into components parallel and orthogonal to $\mathbf{x}$. The matrix $\mathbf{I}_d - \frac{\mathbf{x}\mathbf{x}^T}{d\,r(\mathbf{x})^2}$ possesses two distinct types of eigenvectors:

1. Any vector $\mathbf{v}_\perp$ orthogonal to $\mathbf{x}$ (i.e., $\mathbf{x}^T\mathbf{v}_\perp = 0$) is an eigenvector with eigenvalue $\lambda_\perp = 1$.

2. The vector $\mathbf{v}_\| = \mathbf{x}$ itself is an eigenvector, yielding:

$$\begin{aligned}
\left(\mathbf{I}_d - \frac{\mathbf{x}\mathbf{x}^T}{d\,r(\mathbf{x})^2}\right)\mathbf{x} &= \mathbf{x} - \frac{\|\mathbf{x}\|_2^2}{d\,r(\mathbf{x})^2}\mathbf{x} \\
&= \left(1 - \frac{\|\mathbf{x}\|_2^2}{\|\mathbf{x}\|_2^2 + d\epsilon}\right)\mathbf{x} = \left(\frac{d\epsilon}{\|\mathbf{x}\|_2^2 + d\epsilon}\right)\mathbf{x}.
\end{aligned}$$

Thus, the corresponding eigenvalue is $\lambda_\| = \frac{d\epsilon}{\|\mathbf{x}\|_2^2 + d\epsilon}$.

Since $\epsilon > 0$, the eigenvalues lie strictly within the interval $(0, 1]$, and the maximum eigenvalue is exactly 1 (associated with the orthogonal subspace). Therefore, the spectral norm of the bracketed matrix is exactly 1, which implies:

$$\|J_y(\mathbf{x})\|_2 = \frac{1}{r(\mathbf{x})}\left\|\mathbf{I}_d - \frac{\mathbf{x}\mathbf{x}^T}{d\,r(\mathbf{x})^2}\right\|_2 = \frac{1}{r(\mathbf{x})}.$$

Using the absolute lower bound $r(\mathbf{x}) = \sqrt{\frac{1}{d}\|\mathbf{x}\|_2^2 + \epsilon} \geq \sqrt{\epsilon}$, the spectral norm of the Jacobian is uniformly bounded across the entire domain:

$$\|J_y(\mathbf{x})\|_2 \leq \frac{1}{\sqrt{\epsilon}}.$$

By the mean value theorem for vector-valued functions, this uniform bound on the Jacobian implies that $y(\mathbf{x})$ is globally $(1/\sqrt{\epsilon})$-Lipschitz continuous:

$$\|y(\mathbf{x}_1) - y(\mathbf{x}_2)\|_2 \leq \frac{1}{\sqrt{\epsilon}}\|\mathbf{x}_1 - \mathbf{x}_2\|_2.$$

Finally, scaling by the weight matrix parameter yields:

$$\|\text{RMSNorm}(\mathbf{x}_1) - \text{RMSNorm}(\mathbf{x}_2)\|_2 \leq \frac{\|\mathbf{g}\|_\infty}{\sqrt{\epsilon}}\|\mathbf{x}_1 - \mathbf{x}_2\|_2,$$

which completes the proof. $\square$

**Lemma A.4** (Bounded Magnitude of RMSNorm). *For any* $\mathbf{x} \in \mathbb{R}^d \setminus \{\mathbf{0}\}$, *the Root Mean Square Layer Normalization satisfies the following magnitude bound:*

$$\|\text{RMSNorm}(\mathbf{x})\|_2 \leq \sqrt{d} \cdot \|\mathbf{g}\|_\infty,$$

*where* $\|\mathbf{g}\|_\infty = \max_i |g_i^l|$. *Specifically, when* $\mathbf{G}^l = \mathbf{I}$ *and* $\epsilon = 0$, *we have* $\|\text{RMSNorm}(\mathbf{x})\|_2 = \sqrt{d}$.

*Proof.* Let $\bar{\mathbf{x}} = \frac{\mathbf{x}}{\sqrt{\frac{1}{d}\|\mathbf{x}\|_2^2 + \epsilon}}$. Its $\ell_2$-norm satisfies:

$$\|\bar{\mathbf{x}}\|_2 = \sqrt{\frac{d \cdot \|\mathbf{x}\|_2^2}{\|\mathbf{x}\|_2^2 + d\epsilon}} \leq \sqrt{d},$$

where the inequality holds holds with equality if and only if $\epsilon = 0$. Since $\text{RMSNorm}(\mathbf{x}) = \mathbf{G}^l\bar{\mathbf{x}}$ and $\mathbf{G}^l = \text{diag}(\mathbf{g}^l)$, by the compatibility of the matrix spectral norm and vector $\ell_2$-norm, we immediately obtain:

$$\|\text{RMSNorm}(\mathbf{x})\|_2 \leq \|\mathbf{G}^l\|_2\|\bar{\mathbf{x}}\|_2 \leq \|\mathbf{g}\|_\infty \cdot \sqrt{d}.$$

$\square$

After the RMSNorm, there are three projection matrix, and the bound of difference between output can be easily proved.

**Lemma A.5** (Lipschitz Continuity of Linear Transformation). *Let $\mathbf{x}_1, \mathbf{x}_2 \in \mathbb{R}^d$ and let $\mathbf{W} \in \mathbb{R}^{d \times d}$ be a matrix. Then the Euclidean distance between the transformed vectors is bounded by*

$$\|\mathbf{x}_1 \mathbf{W} - \mathbf{x}_2 \mathbf{W}\|_2 \leq \|\mathbf{W}\|_2 \cdot \|\mathbf{x}_1 - \mathbf{x}_2\|_2,$$

*where $\|\mathbf{W}\|_2$ denotes the spectral norm (largest singular value) of $W$.*

*Proof.* By linearity of matrix multiplication, we have

$$\mathbf{x}_1 \mathbf{W} - \mathbf{x}_2 \mathbf{W} = (\mathbf{x}_1 - \mathbf{x}_2)\mathbf{W}.$$

Taking the Euclidean norm on both sides yields

$$\|\mathbf{x}_1 \mathbf{W} - \mathbf{x}_2 \mathbf{W}\|_2 = \|(\mathbf{x}_1 - \mathbf{x}_2)\mathbf{W}\|_2.$$

By the submultiplicativity property of induced matrix norms, we obtain

$$\|(\mathbf{x}_1 - \mathbf{x}_2)\mathbf{W}\|_2 \leq \|\mathbf{x}_1 - \mathbf{x}_2\|_2 \cdot \|\mathbf{W}\|_2.$$

Combining the above inequalities completes the proof. $\square$

Next component is the attention computation, and we will first derive the bound on difference between inner product before being applied softmax.

**Lemma A.6** (Stability of Inner Product). *Let $\mathbf{q}_1, \mathbf{q}_2, \mathbf{k}_1, \mathbf{k}_2 \in \mathbb{R}^d$. The difference between the corresponding inner products is bounded by*

$$\left| \mathbf{q}_1^\top \mathbf{k}_1 - \mathbf{q}_2^\top \mathbf{k}_2 \right| \leq \varepsilon_{qk} (\|\mathbf{q}_1 - \mathbf{q}_2\|_2 + \|\mathbf{k}_1 - \mathbf{k}_2\|_2).$$

*where $\varepsilon_{qk} = \max(\|\mathbf{q}_1\|_2, \|\mathbf{q}_2\|_2, \|\mathbf{k}_1\|_2, \|\mathbf{k}_2\|_2)$*

*Proof.* We decompose the difference as follows:

$$\mathbf{q}_1^\top \mathbf{k}_1 - \mathbf{q}_2^\top \mathbf{k}_2 = \mathbf{q}_1^\top \mathbf{k}_1 - \mathbf{q}_2^\top \mathbf{k}_1 + \mathbf{q}_2^\top \mathbf{k}_1 - \mathbf{q}_2^\top \mathbf{k}_2.$$

Rearranging terms gives

$$= (\mathbf{q}_1 - \mathbf{q}_2)^\top \mathbf{k}_1 + \mathbf{q}_2^\top (\mathbf{k}_1 - \mathbf{k}_2).$$

Taking absolute values and applying the triangle inequality yields

$$\left| \mathbf{q}_1^\top \mathbf{k}_1 - \mathbf{q}_2^\top \mathbf{k}_2 \right| \leq \left| (\mathbf{q}_1 - \mathbf{q}_2)^\top \mathbf{k}_1 \right| + \left| \mathbf{q}_2^\top (\mathbf{k}_1 - \mathbf{k}_2) \right|.$$

Using the Cauchy–Schwarz inequality on each term, we obtain

$$\leq \|\mathbf{q}_1 - \mathbf{q}_2\|_2 \|\mathbf{k}_1\|_2 + \|\mathbf{q}_2\|_2 \|\mathbf{k}_1 - \mathbf{k}_2\|_2.$$

Applying the $\varepsilon_{qk}$ completes the proof:

$$\leq \varepsilon_{qk}(\|\mathbf{q}_1 - \mathbf{q}_2\|_2 + \|\mathbf{k}_1 - \mathbf{k}_2\|_2).$$

$\square$

Then, we will show that after being applied softmax function and weighted average the value, there is still a bound between two outputs.

**Lemma A.7** (Softmax Stability). *Let $\mathbf{p}_1, \mathbf{p}_2 \in \mathbb{R}^\eta$ be two vectors of pre-softmax scores, and define*

$$\mathbf{a}_1 = \mathrm{softmax}(\mathbf{p}_1), \quad \mathbf{a}_2 = \mathrm{softmax}(\mathbf{p}_2),$$

*where*

$$\mathrm{softmax}(\mathbf{p})_i = \frac{e^{p_i}}{\sum_{j=1}^{n} e^{p_j}}.$$

*Then the softmax outputs satisfy*

$$\|\mathbf{a}_1 - \mathbf{a}_2\|_2 \leq \frac{1}{2}\|\mathbf{p}_1 - \mathbf{p}_2\|_2.$$

*Proof.* The Jacobian matrix of the softmax function at $\mathbf{p}$ is given by

$$J_{\mathrm{softmax}}(\mathbf{p}) = \nabla \mathrm{softmax}(\mathbf{p}) = \mathrm{diag}(\mathbf{a}) - \mathbf{a}\mathbf{a}^\top,$$

where $\mathbf{a} = \mathrm{softmax}(\mathbf{p})$.

It is known that the spectral norm of this Jacobian is uniformly bounded:

$$\|J(\mathbf{p})\|_2 \leq \frac{1}{2}$$

for all $\mathbf{p} \in \mathbb{R}^\eta$.

By the mean value theorem for vector-valued functions, this uniform bound on the Jacobian implies that $\mathrm{softmax}(\mathbf{p})$ is globally ($\frac{1}{2}$)-Lipschitz continuous:

$$\|\mathbf{a}_1 - \mathbf{a}_2\|_2 \leq \frac{1}{2}\|\mathbf{p}_1 - \mathbf{p}_2\|_2.$$

which completes the proof. $\square$

**Lemma A.8** (Stability of Attention Value Aggregation ). *Let $\mathbf{a}_1, \mathbf{a}_2 \in \mathbb{R}^\eta$ be attention weight vectors and $\mathbf{V}_1, \mathbf{V}_2 \in \mathbb{R}^{\eta \times d}$ be value matrices, we assume any row vector $\mathbf{v}_{1,i}, \mathbf{v}_{2,i}$ in $\mathbf{V}_1, \mathbf{V}_2$ satisfy:*

$$\|\mathbf{v}_{1,i}\|_2 \leq \varepsilon_v, \|\mathbf{v}_{2,i}\|_2 \leq \varepsilon_v$$

*Then the attention outputs satisfy the following bound:*

$$\|\mathbf{a}_1 \mathbf{V}_1 - \mathbf{a}_2 \mathbf{V}_2\|_2$$
$$\leq \varepsilon_v \sqrt{\eta} \|\mathbf{a}_1 - \mathbf{a}_2\|_2 + \sqrt{\sum_{i=1}^{\eta} \|\mathbf{v}_{1,i} - \mathbf{v}_{2,i}\|_2^2}.$$

*Proof.* We decompose the difference as

$$\mathbf{a}_1 \mathbf{V}_1 - \mathbf{a}_2 \mathbf{V}_2 = (\mathbf{a}_1 - \mathbf{a}_2)\mathbf{V}_1 + \mathbf{a}_2(\mathbf{V}_1 - \mathbf{V}_2).$$

Applying the triangle inequality yields

$$\|\mathbf{a}_1 \mathbf{V}_1 - \mathbf{a}_2 \mathbf{V}_2\|_2 \leq \|(\mathbf{a}_1 - \mathbf{a}_2)\mathbf{V}_1\|_2 + \|\mathbf{a}_2(\mathbf{V}_1 - \mathbf{V}_2)\|_2.$$

**Bounding the first term.** Using the column expansion and the bounded norm assumption,

$$(\mathbf{a}_1 - \mathbf{a}_2)\mathbf{V}_1 = \sum_{i=1}^{\eta}(a_{1,i} - a_{2,i})\mathbf{v}_{1,i}.$$

Thus, with the norm assumption

$$\|(\mathbf{a}_1 - \mathbf{a}_2)\mathbf{V}_1\|_2 \leq \sum_{i=1}^{\eta}|a_{1,i} - a_{2,i}|\,\|\mathbf{v}_{1,i}\|_2 \leq \varepsilon_v\|\mathbf{a}_1 - \mathbf{a}_2\|_1.$$

Applying the inequality $\|\mathbf{x}\|_1 \leq \sqrt{\eta}\|\mathbf{x}\|_2$ gives

$$\|(\mathbf{a}_1 - \mathbf{a}_2)\mathbf{V}_1\|_2 \leq \varepsilon_v\sqrt{\eta}\|\mathbf{a}_1 - \mathbf{a}_2\|_2$$

**Bounding the second term.** Similarly,

$$\mathbf{a}_2(\mathbf{V}_1 - \mathbf{V}_2) = \sum_{i=1}^{\eta}a_{2,i}(\mathbf{v}_{1,i} - \mathbf{v}_{2,i}).$$

Applying the Cauchy–Schwarz inequality,

$$\|\mathbf{a}_2(\mathbf{V}_1 - \mathbf{V}_2)\|_2 \leq \sqrt{\sum_{i=1}^{\eta}a_{2,i}^2}\sqrt{\sum_{i=1}^{\eta}\|\mathbf{v}_{1,i} - \mathbf{v}_{2,i}\|_2^2}.$$

Since $\mathbf{a}_2$ is a probability vector, we have $\|\mathbf{a}_2\|_2 \leq 1$, therefore

$$\|\mathbf{a}_2(\mathbf{V}_1 - \mathbf{V}_2)\|_2 \leq \sqrt{\sum_{i=1}^{\eta}\|\mathbf{v}_{1,i} - \mathbf{v}_{2,i}\|_2^2}.$$

**Combining the bounds.** Summing the two terms yields

$$\|\mathbf{a}_1\mathbf{V}_1 - \mathbf{a}_2\mathbf{V}_2\|_2$$
$$\leq \varepsilon_v\sqrt{\eta}\|\mathbf{a}_1 - \mathbf{a}_2\|_2 + \sqrt{\sum_{i=1}^{\eta}\|\mathbf{v}_{1,i} - \mathbf{v}_{2,i}\|_2^2}.$$

This completes the proof. □

The final component of a transformer block is feed-forward network.

**Lemma A.9** (Lipschitz Continuity of SwiGLU Feed-Forward Network). *Let $\mathbf{x}_1, \mathbf{x}_2 \in \mathbb{R}^d$ be two position-wise row vectors from the normalized intermediate representation matrix $\tilde{\mathbf{U}}^l$. Consider the feed-forward network sublayer operation:*

$$\mathrm{FFN}(\mathbf{x}) = \mathrm{SwiGLU}(\mathbf{x}\mathbf{W}_1^l + \mathbf{b}_1^l)\mathbf{W}_2^l + \mathbf{b}_2^l,$$

*where $\mathbf{W}_1^l, \mathbf{W}_2^l \in \mathbb{R}^{d\times d}$, and the non-linear activation $\sigma(\cdot)$ is specified as the SwiGLU function:*

$$\mathrm{SwiGLU}(\mathbf{z}) = \mathrm{Swish}(\mathbf{z}_1) \odot \mathbf{z}_2,$$

*with $\mathbf{z} = [\mathbf{z}_1, \mathbf{z}_2] \in \mathbb{R}^d$ being an equal split of channels $(\mathbf{z}_1, \mathbf{z}_2 \in \mathbb{R}^{d/2})$ and $\mathrm{Swish}(t) = t \cdot \sigma(t)$.*

*Then the position-wise output difference is bounded by:*

$$\|\mathrm{FFN}(\mathbf{x}_1) - \mathrm{FFN}(\mathbf{x}_2)\|_2$$
$$\leq \|\mathbf{W}_2^l\|_2 \cdot L_{\mathrm{SwiGLU}} \cdot \|\mathbf{W}_1^l\|_2 \cdot \|\mathbf{x}_1 - \mathbf{x}_2\|_2,$$

*where $L_{\mathrm{SwiGLU}}$ is the local Lipschitz constant of the SwiGLU activation evaluated over the bounded domain of the intermediate representations.*

*Proof.* We decompose the position-wise FFN mapping into three sequential operations:

$$\mathbf{h}(\mathbf{x}) = \mathbf{x}\mathbf{W}_1^l + \mathbf{b}_1^l,$$
$$\mathbf{g}(\mathbf{h}) = \mathrm{SwiGLU}(\mathbf{h}),$$
$$\mathbf{y}(\mathbf{g}) = \mathbf{g}\mathbf{W}_2^l + \mathbf{b}_2^l.$$

**Step 1: Linear projection and domain boundedness.**

By Lemma A.5 and the property of the matrix spectral norm, the difference after the first linear projection satisfies:

$$\|\mathbf{h}(\mathbf{x}_1) - \mathbf{h}(\mathbf{x}_2)\|_2 = \|(\mathbf{x}_1 - \mathbf{x}_2)\mathbf{W}_1^l\|_2$$
$$\leq \|\mathbf{W}_1^l\|_2\|\mathbf{x}_1 - \mathbf{x}_2\|_2.$$

Furthermore, since the inputs $\mathbf{x}_1, \mathbf{x}_2$ are rows of the normalized matrix $\tilde{\mathbf{U}}^l = \mathrm{RMSNorm}_2(\mathbf{U}^l)$, it follows directly from Lemma A.4 that their magnitudes are globally bounded:

$$\|\mathbf{x}_i\|_2 \leq \sqrt{d} \cdot \|\mathbf{g}\|_\infty =: M_x, \quad \text{for } i \in \{1, 2\}.$$

Consequently, the intermediate representation $\mathbf{h}_i = \mathbf{h}(\mathbf{x}_i)$ is restricted to a compact (bounded and closed) domain $\Omega \subset \mathbb{R}^d$, where the norm of each element is bounded by:

$$\|\mathbf{h}_i\|_2 \leq M_x\|\mathbf{W}_1^l\|_2 + \|\mathbf{b}_1^l\|_2 =: M_h.$$

**Step 2: SwiGLU nonlinearity on a compact domain.**

The SwiGLU activation function is continuously differentiable $(\mathcal{C}^1)$, implying its Jacobian matrix $\mathcal{J}_{\mathrm{SwiGLU}}$ is continuous. While its Jacobian norm is unbounded on the entire space $\mathbb{R}^d$, it is strictly bounded on any compact subset. Since $\mathbf{h}_1, \mathbf{h}_2 \in \Omega$, by the Mean Value Theorem, there exists a well-defined local Lipschitz constant $L_{\mathrm{SwiGLU}} = \sup_{\mathbf{h}\in\Omega}\|\mathcal{J}_{\mathrm{SwiGLU}}(\mathbf{h})\|_2 < \infty$ such that:

$$\|\mathbf{g}(\mathbf{h}_1) - \mathbf{g}(\mathbf{h}_2)\|_2 \leq L_{\mathrm{SwiGLU}}\|\mathbf{h}_1 - \mathbf{h}_2\|_2.$$

**Step 3: Output projection.**

Again, by Lemma A.5, the final linear layer yields:

$$\|\mathbf{y}(\mathbf{g}_1) - \mathbf{y}(\mathbf{g}_2)\|_2 = \|(\mathbf{g}_1 - \mathbf{g}_2)\mathbf{W}_2^l\|_2$$
$$\leq \|\mathbf{W}_2^l\|_2 \|\mathbf{g}_1 - \mathbf{g}_2\|_2.$$

**Step 4: Combine all bounds.**

Combining the inequalities from Steps 1 to 3 via sequential substitution, we obtain:

$$\|\text{FFN}(\mathbf{x}_1) - \text{FFN}(\mathbf{x}_2)\|_2$$
$$\leq \|\mathbf{W}_2^l\|_2 \cdot L_{\text{SwiGLU}} \cdot \|\mathbf{W}_1^l\|_2 \cdot \|\mathbf{x}_1 - \mathbf{x}_2\|_2.$$

$\square$

Now, we have constructed the bound for each components in a transformer block, and the last bound we need is for the language model head.

**Lemma A.10.** *(Stability of LM Head) Let $\mathbf{x}_1, \mathbf{x}_2 \in \mathbb{R}^d$ be two embeddings. Let $P_1$ and $P_2$ be the probability distributions produced by a Language Model head with weight matrix $\mathbf{W} \in \mathbb{R}^{V \times d}$ and bias $\mathbf{b} \in \mathbb{R}^V$, such that $P_i = softmax(\mathbf{W}\mathbf{x}_i + \mathbf{b})$.*

*The probability that the same token is sampled from both distributions, $P(same) = \sum_{k=1}^V P_1(k)P_2(k)$, is lower-bounded by:*

$$P(same) \geq \sum_{k=1}^V P_1(k)^2 - \max_k \|\mathbf{w}_k\|_2 \cdot \|\mathbf{x}_1 - \mathbf{x}_2\|_2$$

*where $\mathbf{w}_k$ is the k-th row of $\mathbf{W}$.*

*Proof.* We prove this lemma through following steps

**Step 1: Define the Logits:**

Let $\mathbf{z}_1 = \mathbf{W}\mathbf{x}_1 + \mathbf{b}$ and $\mathbf{z}_2 = \mathbf{W}\mathbf{x}_2 + \mathbf{b}$. The difference in logits for any token $k$ is:

$$|z_{1,k} - z_{2,k}| = |\mathbf{w}_k^\top (\mathbf{x}_1 - \mathbf{x}_2)| \leq \|\mathbf{w}_k\|_2 \|\mathbf{x}_1 - \mathbf{x}_2\|_2$$

**Step 2: Lipschitz Property of Softmax:**

The Softmax function $\sigma(\mathbf{z})$ is 1-Lipschitz with respect to the $L_\infty$ norm on the logits and the $L_1$ norm on the output probabilities. Specifically, the Total Variation distance $d_{TV}(P_1, P_2) = \frac{1}{2}\sum_k |P_1(k) - P_2(k)|$ satisfies:

$$d_{TV}(P_1, P_2) \leq \max_k |z_{1,k} - z_{2,k}|$$
$$\leq (\max_k \|\mathbf{w}_k\|_2)\|\mathbf{x}_1 - \mathbf{x}_2\|_2 \quad (3)$$

**Step 3: Relating to Collision Probability:**

Let $\epsilon = d_{TV}(P_1, P_2)$. We can write $P_2(k) = P_1(k) + \delta_k$, where $\sum \delta_k = 0$ and $\frac{1}{2}\sum |\delta_k| = \epsilon$. The probability of sampling the same token is:

$$P(\text{same})$$
$$= \sum_k P_1(k)P_2(k)$$
$$= \sum_k P_1(k)(P_1(k) + \delta_k)$$
$$= \sum_k P_1(k)^2 + \sum_k P_1(k)\delta_k$$

**Step 4: Bounding the Error Term:**

Since $\sum_k \delta_k = 0$, for any constant $C$, we have $\sum_k P_1(k)\delta_k = \sum_k (P_1(k) - C)\delta_k$. By choosing $C$ as the midpoint of the probability range, i.e.,

$$C = \frac{\max_j P_1(j) + \min_j P_1(j)}{2},$$

we can bound the absolute error term using Hölder's inequality:

$$\left|\sum_k P_1(k)\delta_k\right| = \left|\sum_k (P_1(k) - C)\delta_k\right|$$
$$\leq \max_k |P_1(k) - C| \sum_k |\delta_k|$$
$$= \left(\frac{\max_j P_1(j) - \min_j P_1(j)}{2}\right) \cdot (2\epsilon)$$
$$= \epsilon \cdot \text{range}(P_1) \leq \epsilon$$

where $\text{range}(P_1) = \max_j P_1(j) - \min_j P_1(j)$. Since $P_1$ is a valid probability distribution, $P_1(k) \in [0, 1]$, which implies $\text{range}(P_1) \leq 1$.

**Conclusion**

Substituting $\epsilon \leq (\max_k \|\mathbf{w}_k\|_2)\|\mathbf{x}_1 - \mathbf{x}_2\|_2$:

$$P(\text{same}) \geq \sum_k P_1(k)^2 - (\max_k \|\mathbf{w}_k\|_2) \cdot \|\mathbf{x}_1 - \mathbf{x}_2\|_2$$

$\square$

Finally, we can prove the correlation in this simplified case.

**Theorem A.11** (Correlation in Simplified Case). *Suppose $R_t$ is the final reward of generated sequence from state $s_t$. Assume hidden states of $s_1, s_2$ are $\mathbf{X}_1^l, \mathbf{X}_2^l \in \mathbb{R}^{\eta \times d}$. If hidden states of $s_1, s_2$ fulfill Assumption A.1 A.2 and following equation:*

$$i = \arg\min_{j \in \{1,\ldots,\eta\}} \|\mathbf{x}_{1,i}^l - \mathbf{x}_{2,j}^l\|_2, \quad \forall i \in \{1,\ldots,\eta\}.$$

*Then, the probability that $R_1 = R_2$ is correlated with the MinDistance between the hidden states:*

$$P(R_1 = R_2) \propto \frac{1}{\text{MD}(\mathbf{X}_1^l, \mathbf{X}_2^l)}, \quad \forall l \in [1, L]$$

*Proof.* Suppose we continue the generation of $s_1$ and $s_2$ until $(T-1)$-th state located at the token index $\ell$. The final answer will be output as $(T-1)$-th action. Then the enlongated hidden states at 0-th layer are $\bar{\mathbf{X}}_1^0 = (\mathbf{x}_{1,1}^0, \mathbf{x}_{1,2}^0, ..., \mathbf{x}_{1,\eta}^0, \mathbf{x}_{1,\eta+1}^0, ...\mathbf{x}_{1,\ell}^0)$ and $\bar{\mathbf{X}}_2^0 = (\mathbf{x}_{2,1}^0, \mathbf{x}_{2,2}^0, ..., \mathbf{x}_{2,\eta}^0, \mathbf{x}_{2,\eta+1}^0, ...\mathbf{x}_{2,\ell}^0)$. According to Assumption A.1, the final answer will be derived from the forward result of $\mathbf{x}_{1,\ell}^0$ and $\mathbf{x}_{2,\ell}^0$. We can focus on the difference between them along the forward pass. We still begins by ignoring layer superscript $l$ to derive a general formula across layers.

Besides $\mathbf{x}_{1,\ell}$ and $\mathbf{x}_{2,\ell}$, let $\mathbf{x}_{1,i}$ and $\mathbf{x}_{2,i}$ be any hidden state from $\bar{\mathbf{X}}_1$ and $\bar{\mathbf{X}}_2$. In casual attention mechanism, we calculate:

$$\begin{aligned}
\mathbf{q}_{1,\ell} &= \mathbf{x}_{1,\ell}\mathbf{W}_Q, \quad \mathbf{q}_{2,\ell} = \mathbf{x}_{2,\ell}\mathbf{W}_Q \\
\mathbf{k}_{1,i} &= \mathbf{x}_{1,i}\mathbf{W}_K, \quad \mathbf{k}_{2,i} = \mathbf{x}_{2,i}\mathbf{W}_K \\
\mathbf{v}_{1,i} &= \mathbf{x}_{1,i}\mathbf{W}_V, \quad \mathbf{v}_{2,i} = \mathbf{x}_{2,i}\mathbf{W}_V
\end{aligned}$$

By applying Lemma A.3 and Lemma A.5, the difference between corresponding query and key can be bounded:

$$\|\mathbf{q}_{1,\ell} - \mathbf{q}_{2,\ell}\|_2 \le \frac{\|\mathbf{g}_1\|_\infty}{\sqrt{\epsilon_{RMS1}}}\|\mathbf{W}_Q\|_2\|\mathbf{x}_{1,\ell} - \mathbf{x}_{2,\ell}\|_2$$

$$\|\mathbf{k}_{1,i} - \mathbf{k}_{2,i}\|_2 \le \frac{\|\mathbf{g}_1\|_\infty}{\sqrt{\epsilon_{RMS1}}}\|\mathbf{W}_K\|_2\|\mathbf{x}_{1,i} - \mathbf{x}_{2,i}\|_2$$

Since the attention score is based on inner product $\mathbf{q}_{1,\ell}^T\mathbf{k}_{1,i}$ and $\mathbf{q}_{2,\ell}^T\mathbf{k}_{2,i}$, we apply Lemma A.6 to derive the bound of their difference:

$$\begin{aligned}
\|&\frac{\mathbf{q}_{1,\ell}^T\mathbf{k}_{1,i}}{\sqrt{d}} - \frac{\mathbf{q}_{2,\ell}^T\mathbf{k}_{2,i}}{\sqrt{d}}\|_2 \\
&\le \frac{\|\mathbf{g}_1\|_\infty}{\sqrt{d\epsilon_{RMS1}}}\varepsilon_{qk}(\|\mathbf{W}_Q\|_2\|\mathbf{x}_{1,\ell} - \mathbf{x}_{2,\ell}\|_2 \\
&\quad + \|\mathbf{W}_K\|_2\|\mathbf{x}_{1,i} - \mathbf{x}_{2,i}\|_2)
\end{aligned}$$

By applying Lemma A.4, $\varepsilon_{qk}$ can be bounded by:

$$\varepsilon_{qk} \le \sqrt{d} \cdot \|\mathbf{g}_1\|_\infty \cdot \max(\|\mathbf{W}_Q\|_2, \|\mathbf{W}_K\|_2)$$

Before being applied softmax function, for simplicity, let's denote the bound of difference between their dot product score in $i$-th position as

$$\begin{aligned}
\|&\frac{\mathbf{q}_{1,\ell}^T\mathbf{k}_{1,i}}{\sqrt{d}} - \frac{\mathbf{q}_{2,\ell}^T\mathbf{k}_{2,i}}{\sqrt{d}}\|_2 \\
&\le C(\|\mathbf{W}_Q\|_2\|\mathbf{x}_{1,\ell} - \mathbf{x}_{2,\ell}\|_2 + \|\mathbf{W}_K\|_2\|\mathbf{x}_{1,i} - \mathbf{x}_{2,i}\|_2) \\
&= B_{p,i}
\end{aligned}$$

where

$$C = \frac{\|\mathbf{g}_1\|_\infty \cdot \|\mathbf{g}_1\|_\infty}{\sqrt{\epsilon_{RMS1}}} \cdot \max(\|\mathbf{W}_Q\|_2, \|\mathbf{W}_K\|_2).$$

We can apply Lemma A.7 to derive the bound for difference between attention score vector after softmax:

$$\begin{aligned}
\|\mathbf{a}_1 - \mathbf{a}_2\|_2 &\le \frac{1}{2}\left(\sum_{i=1}^{\ell} B_{p,i}^2\right)^{1/2} \\
&\le \frac{C}{2}(\ell\|\mathbf{W}_Q\|_2\|\mathbf{x}_{1,\ell} - \mathbf{x}_{2,\ell}\|_2 \\
&\quad + \|\mathbf{W}_K\|_2\sum_{j=1}^{\eta}\|\mathbf{x}_{1,j} - \mathbf{x}_{2,j}\|_2 \\
&\quad + \|\mathbf{W}_K\|_2\sum_{j=\eta+1}^{\ell}\|\mathbf{x}_{1,j} - \mathbf{x}_{2,j}\|_2) \\
&= B_a
\end{aligned}$$

where $\sum_{j=1}^{\eta}\|\mathbf{x}_{1,j} - \mathbf{x}_{2,j}\|_2 = \text{MD}(\mathbf{X}_1, \mathbf{X}_2)$ in this simplified settings. $\sum_{j=\eta+1}^{\ell}\|\mathbf{x}_{1,j} - \mathbf{x}_{2,j}\|_2$ is from the stochastically generated tokens, which is difficult to measure, and we will substitute it with $\sum_{j=\eta+1}^{\ell}\|\mathbf{x}_{1,j} - \mathbf{x}_{2,j}\|_2 \le 2(\ell - \eta)\sqrt{d} \cdot \|\mathbf{g}_1\|_\infty$ by applying Lemma A.4.

Then, by applying Lemma A.8, the difference between the attention output $\|\mathbf{h}_{1,\ell} - \mathbf{h}_{2,\ell}\|_2$ is bounded by

$$\begin{aligned}
\|\mathbf{h}&_{1,\ell} - \mathbf{h}_{2,\ell}\|_2 \\
&\le \varepsilon_v\sqrt{\ell}B_a + \sqrt{\sum_{j=1}^{\ell}(\frac{\|\mathbf{g}_1\|_\infty}{\sqrt{\epsilon_{RMS1}}}\|\mathbf{W}_V\|_2\|\mathbf{x}_{1,j} - \mathbf{x}_{2,j}\|_2)^2} \\
&\le \varepsilon_v\sqrt{\ell}B_a + \frac{\|\mathbf{g}_1\|_\infty}{\sqrt{\epsilon_{RMS1}}}\|\mathbf{W}_V\|_2(\text{MD}(\mathbf{X}_1, \mathbf{X}_2) \\
&\quad + 2(\ell - \eta)\sqrt{d} \cdot \|\mathbf{g}_1\|_\infty) \\
&= B_h
\end{aligned}$$

where

$$\varepsilon_v \le \sqrt{d} \cdot \|\mathbf{g}_1\|_\infty \cdot \|\mathbf{W}_V\|_2$$

After the output projection matrix and residual connection, the bound of difference after the casual self-attention block is:

$$B_u = \|\mathbf{W}_O\|_2 B_h + \|\mathbf{x}_{1,\ell} - \mathbf{x}_{2,\ell}\|_2$$

The last part of a Transformer block is FFN with SwiGLU as activation function. Apply Lemma A.9, Lemma A.3, and add the residual connection, the difference is bounded by

$$B_x = B_u + \frac{\|\mathbf{g}_2\|_\infty}{\sqrt{d\epsilon_{RMS2}}}\|\mathbf{W}_2\|_2 \cdot L_{\text{SwiGLU}} \cdot \|\mathbf{W}_1\|_2 \cdot B_u$$

Now, we introduce the notation for layer $l$ back to the expression since we have finish the general derivation. To

simplify the expressions, for each layer, let's define following constant:

$$C_\ell^l = \|\mathbf{W}_O^l\|_2 \cdot \frac{\|\mathbf{g}_1^l\|_\infty^3 \ell\sqrt{d\ell}}{2\sqrt{\epsilon_{RMS1}^l}} \cdot \|\mathbf{W}_V^l\|_2 \cdot \|\mathbf{W}_Q^l\|_2$$
$$\cdot \max(\|\mathbf{W}_Q^l\|_2, \|\mathbf{W}_K^l\|_2) + 1$$

$$C_{MLP}^l = 1 + \frac{\|\mathbf{g}_2^l\|_\infty}{\sqrt{d\epsilon_{RMS2}^l}}\|\mathbf{W}_2^l\|_2 \cdot L_{\text{SwiGLU}} \cdot \|\mathbf{W}_1^l\|_2$$

$$C_{MD}^l = \|\mathbf{W}_O^l\|_2 \cdot \frac{\|\mathbf{g}_1^l\|_\infty}{\sqrt{\epsilon_{RMS1}^l}} \cdot \|\mathbf{W}_V^l\|_2$$
$$\cdot (\frac{\|\mathbf{g}_1^l\|_\infty^2\sqrt{d\ell}}{2} \max(\|\mathbf{W}_Q^l\|_2, \|\mathbf{W}_K^l\|_2)\|\mathbf{W}_K^l\|_2 + 1)$$

For $\mathbf{x}_{1,\ell}^{l-1}$ and $\mathbf{x}_{2,\ell}^{l-1}$, after one transformer block, the output satisfy following inequality

$$\|\mathbf{x}_{1,\ell}^l - \mathbf{x}_{2,\ell}^l\|_2 \leq C_\ell^l C_{MLP}^l \|\mathbf{x}_{1,\ell}^{l-1} - \mathbf{x}_{2,\ell}^{l-1}\|_2$$
$$+ C_{MD}^l C_{MLP}^l \text{MD}(\mathbf{X}_1^{l-1}, \mathbf{X}_2^{l-1})$$
$$+ C_{MD}^l C_{MLP}^l 2(\ell - \eta)\sqrt{d} \cdot \|\mathbf{g}_1^l\|_\infty$$

Suppose there is in total $L$ layers in the LLM. For $\ell$-th token of two states $s_1$ and $s_2$, begin from their embedding $\mathbf{x}_{1,\ell}^0$ and $\mathbf{x}_{2,\ell}^0$, their difference in the last layer is:

$$\|\mathbf{x}_{1,\ell}^L - \mathbf{x}_{2,\ell}^L\|_2 \leq$$
$$\left(\prod_{l=1}^L C_\ell^l C_{MLP}^l\right)\|\mathbf{x}_{1,\ell}^0 - \mathbf{x}_{2,\ell}^0\|_2$$
$$+ \sum_{k=1}^L \left(\prod_{l=k+1}^L C_\ell^l C_{MLP}^l\right) C_{MD}^k C_{MLP}^k \text{MD}(\mathbf{X}_1^{k-1}, \mathbf{X}_2^{k-1})$$
$$+ \sum_{k=1}^L \left(\prod_{l=k+1}^L C_\ell^l C_{MLP}^l\right) C_{MD}^k C_{MLP}^k 2(\ell - \eta)\sqrt{d}\|\mathbf{g}_1^{k-1}\|_\infty$$

Suppose $P_1 = \text{softmax}(\mathbf{W}_l\mathbf{x}_{1,\ell} + \mathbf{b}_l)$, where $\mathbf{W}_l$ and $\mathbf{b}_l$ is the weight and bias for language model head. Suppose final reward of $s_1$ is $R_1$ and final reward of $s_2$ is $R_2$. We applied Lemma A.10 to derive

$$P(R_1 = R_2)$$
$$\geq \sum_{j=1}^V P_1(j)^2 - \max_j \|\mathbf{w}_{a,j}\|_2 \cdot \|\mathbf{x}_{1,\ell}^L - \mathbf{x}_{2,\ell}^L\|_2 \quad (4)$$

In this expression, following terms are intractable because

of the stochastic nature of generated output:

$$(\prod_{l=1}^L C_\ell^l C_{MLP}^l)\|\mathbf{x}_{1,\ell}^0 - \mathbf{x}_{2,\ell}^0\|_2$$
$$\sum_{j=1}^V P_1(j)^2$$

and following terms are constant

$$\sum_{k=1}^L (\prod_{l=k+1}^L C_\ell^l C_{MLP}^l)C_{MD}^k C_{MLP}^k 2(\ell - \eta)\sqrt{d}\|\mathbf{g}_1^{k-1}\|_\infty$$
$$\max_j \|\mathbf{w}_{a,j}\|_2$$

The only tractable term is

$$\sum_{k=1}^L \left(\prod_{l=k+1}^L C_\ell^l C_{MLP}^l\right) C_{MD}^k C_{MLP}^k \text{MD}(\mathbf{X}_1^{k-1}, \mathbf{X}_2^{k-1})$$

Therefore, we can conclude that probability of $s_1$ and $s_2$ share the same reward is correlated with *MinDistance* between their hidden states:

$$P(R_1 = R_2) \propto \frac{1}{\text{MD}(\mathbf{X}_1^l, \mathbf{X}_2^l)}, \quad \forall l \in [1, L]$$

$\square$

## A.3. A General Case

In previous section, we only prove the correlation between *MinDistance* of hidden states and final reward in a simple case, where

1. Hidden states of $s_1, s_2$ are $\mathbf{X}_1^l, \mathbf{X}_2^l \in \mathbb{R}^{\eta \times d}$.

2. The equation $i = \arg\min_{j \in \{1,...,\eta\}} \|\mathbf{x}_{1,i}^l - \mathbf{x}_{2,j}^l\|_2$ always holds for $\forall i \in \{1, \ldots, \eta\}$..

3. The final answer of both states follow Assumption A.1 and A.2.

To expand Theorem A.11 to general case, we need to review these three assumptions. Let's begin with the Assumption A.2. Because we can control the generation length by controlling the sampling of output token, we can always ensure generated sequences from two states have certain length and the length $\ell$ can take any value, which means we can safely apply this assumption to the general case.

For the first two assumption, let's assume state $s_1$ has length of $\eta_1$ and $s_2$ has length of $\eta_2$. Without loss of generality,

suppose $\eta_1 \geq \eta_2$, then we can construct a "fake hidden states" using $\mathbf{X}_2^l$:

$$\hat{\mathbf{X}}_2^l = (\hat{\mathbf{x}}_{2,1}^l, \hat{\mathbf{x}}_{2,2}^l, ..., \hat{\mathbf{x}}_{2,\eta_1}^l)$$

such that for $i$-th token in $\hat{\mathbf{X}}_2^l$, it satisfies

$$\hat{\mathbf{x}}_{2,i}^l = \operatorname{argmin}_{\mathbf{x}_{2,j}^l \in \mathbf{X}_2^l} \|\mathbf{x}_{1,i}^l - \mathbf{x}_{2,j}^l\|_2$$

which means

$$\mathrm{MD}(\mathbf{X}_1^l, \mathbf{X}_2^l) = \mathrm{MD}(\mathbf{X}_1^l, \hat{\mathbf{X}}_2^l) = \sum_{i=1}^{\eta_1} \|\mathbf{x}_{1,i}^l - \hat{\mathbf{x}}_{2,i}^l\|_2$$

Then, we can apply the Theorem A.11 to $\mathbf{X}_1^l$ and $\hat{\mathbf{X}}_2^l$ to derive the correlation. Now, the problem becomes, the relation between the final reward of $\hat{s}_2$ and $s_2$. Let's control the generated sequence length of $\hat{s}_2$ and $s_2$ so that the final answer of $\hat{s}_2$ is the forward result of $(\ell + \eta_1 - \eta_2)$-th token and the final answer of $s_2$ is the forward result of $\ell$-th token. Thus, each sequence can divided into two block. First block is "shared", because the first $\eta_1$-th tokens of $\hat{s}_2$ is always from the first $\eta_2$-th tokens of $s_2$. The second block is "generated", because the last $\ell - \eta_1$ tokens of $\hat{s}_2$ and the last $\ell - \eta_2$ tokens of $s_2$ is stochastically generated and intractable. Then, since part of the $\mathbf{Q}^l$, $\mathbf{K}^l$, and $\mathbf{V}^l$ for those two states are the same, by grouping the same parts together, it can be treated as a special pattern during attention computation, which has following property:

**Lemma A.12** (Structured attention score after softmax aggregation). *Let $\mathbf{p}_1 \in \mathbb{R}^\ell$. Construct $\mathbf{p}_2 \in \mathbb{R}^{\ell+\eta_1-\eta_2}$ by two blocks:*

1. *For each $i \in \{1, \ldots, \eta_2\}$, the dot product score $p_{1,i}$ is replicated $\hat{r}_i$ times in the first block of $\mathbf{p}_2$, where $\hat{r}_i$ are nonnegative integers satisfying $\sum_{i=1}^{\eta_2} \hat{r}_i = \eta_1$. (We allow $\hat{r}_i = 0$ for some $i$.)*

2. *For each $i \in \{\eta_2 + 1, \ldots, \ell\}$, the entry $p_{2,\eta_1+i-\eta_2}$ in $\mathbf{p}_2$ equals $p_{1,i} + \delta_i$ for some $\delta_i \in \mathbb{R}$.*

*Define*

$$\mathbf{a}_1 := \operatorname{softmax}(\mathbf{p}_1), \qquad \mathbf{a}_2 := \operatorname{softmax}(\mathbf{p}_2),$$

*and let $\mathbf{a}_2' \in \mathbb{R}^\ell$ be the vector obtained by aggregating the scores of $\mathbf{a}_2$'s first block according to the originating $p_{1,i}$ and placing the aggregated mass in coordinate $i$ (coordinates $i > \eta_2$ remain aligned as described). Define*

$$\kappa_i := \begin{cases} \hat{r}_i, & i = 1, \ldots, \eta_2, \\ e^{\delta_i}, & i = \eta_2 + 1, \ldots, \ell, \end{cases} \qquad \bar{\kappa} := \sum_{i=1}^{\ell} a_{1,i}\, \kappa_i.$$

*Then for every $i \in \{1, \ldots, \ell\}$,*

$$a_{2,i}' = \frac{a_{1,i}\, \kappa_i}{\bar{\kappa}}$$

*(with the convention $a_{2,i}' = 0$ when $\hat{r}_i = 0$), and the distance satisfies*

$$\|\mathbf{a}_1 - \mathbf{a}_2'\|_2^2 = \sum_{i=1}^{\ell} a_{1,i}^2 \left(1 - \frac{\kappa_i}{\bar{\kappa}}\right)^2.$$

*Proof.* Let

$$Z_1 := \sum_{i=1}^{\ell} e^{p_{1,i}}, \qquad Z_2 := \sum_{i=1}^{\ell+\eta_1-\eta_2} e^{p_{2,i}}.$$

Regrouping the terms of $Z_2$ according to the original indices $i = 1, \ldots, \ell$ of $\mathbf{p}_1$ gives

$$Z_2 = \sum_{i=1}^{\ell} e^{p_{1,i}} \kappa_i,$$

because each $i \leq \eta_2$ contributes $\hat{r}_i$ copies of $e^{p_{1,i}}$ and each $i > \eta_2$ contributes $e^{p_{1,i}+\delta_i} = e^{p_{1,i}} e^{\delta_i}$. Hence for $\mathbf{a}_1$ we have $a_{1,i} = e^{p_{1,i}}/Z_1$. The aggregated entry in $\mathbf{a}_2'$ corresponding to index $i$ is

$$a_{2,i}' = \frac{e^{p_{1,i}} \kappa_i}{Z_2}.$$

Define $\bar{\kappa} = \sum_{i=1}^{\ell} a_{1,i} \kappa_i$. Substituting $a_{1,i} = e^{p_{1,i}}/Z_1$ into the sum yields

$$\bar{\kappa} = \sum_{i=1}^{\ell} \frac{e^{p_{1,i}}}{Z_1} \kappa_i = \frac{1}{Z_1} \sum_{i=1}^{\ell} e^{p_{1,i}} \kappa_i = \frac{Z_2}{Z_1}.$$

Thus $Z_2 = Z_1 \bar{q}$ and

$$a_{2,i}' = \frac{e^{p_{1,i}} \kappa_i}{Z_2} = \frac{(e^{p_{1,i}}/Z_1)\kappa_i}{\bar{\kappa}} = \frac{a_{1,i}\kappa_i}{\bar{\kappa}}.$$

If $\kappa_i = 0$ then $a_{2,i}' = 0$ as claimed. The coordinate-wise difference is therefore

$$a_{1,i} - a_{2,i}' = a_{1,i}\left(1 - \frac{\kappa_i}{\bar{\kappa}}\right),$$

and squaring and summing over $j = 1, \ldots, \alpha$ yields

$$\|\mathbf{a}_1 - \mathbf{a}_2'\|_2^2 = \sum_{i=1}^{\ell} a_{1,i}^2 \left(1 - \frac{\kappa_i}{\bar{\kappa}}\right)^2,$$

which completes the proof. $\qquad\square$

We can use Lemma A.12 to build correlation between $\mathbf{X}_2^l$ and $\hat{\mathbf{X}}_2^l$ to prove the general case.

**Theorem A.13** (Correlation in General Case). *Suppose $R_i$ is the final reward of generated sequence from state $s_i$, which fulfill the Assumption A.1. Then, the probability that $R_1 = R_2$ is correlated with the MinDistance between the hidden states:*

$$P(R_1 = R_2) \propto \frac{1}{\mathrm{MD}(\mathbf{X}_1^l, \mathbf{X}_2^l)}, \quad \forall l \in [1, L]$$

*Proof.* Without loss of generality, we assume $s_1$'s hidden states $\mathbf{X}_1^l \in \mathbb{R}^{\eta_1, d}$ and $s_2$'s hidden states $\mathbf{X}_2^l \in \mathbb{R}^{\eta_2, d}$, where $\eta_1 > \eta_2$. To align $\mathbf{X}_1^l$ and $\mathbf{X}_2^l$, we construct a "fake hidden states" for $\mathbf{X}_2^l$:

$$\hat{\mathbf{X}}_2^l = (\hat{\mathbf{x}}_{2,1}^l, \hat{\mathbf{x}}_{2,2}^l, ..., \hat{\mathbf{x}}_{2,\eta_1}^l)$$

such that for $i$-th token in $\hat{\mathbf{X}}_2^l$, it satisfies

$$\hat{\mathbf{x}}_{2,i}^l = \mathrm{argmin}_{\mathbf{x}_{2,j}^l \in \mathbf{X}_2^l} \|\mathbf{x}_{1,i}^l - \mathbf{x}_{2,j}^l\|_2$$

Suppose we continue the generation of $s_1, \hat{s}_2$ until $(\ell + \eta_1 - \eta_2)$-th token and $s_2$ until $\ell$-th token, then the elongated hidden states are

$$\bar{\mathbf{X}}_1^l = (\mathbf{x}_{1,1}^l, \mathbf{x}_{1,2}^l, ..., \mathbf{x}_{1,\eta_1}^l, \mathbf{x}_{1,\eta_1+1}^l, ...\mathbf{x}_{1,\ell+\eta_1-\eta_2}^l)$$
$$\bar{\hat{\mathbf{X}}}_2^l = (\hat{\mathbf{x}}_{2,1}^l, \hat{\mathbf{x}}_{2,2}^l, ..., \hat{\mathbf{x}}_{2,\eta_1}^l, \hat{\mathbf{x}}_{2,\eta_1+1}^l, ...\hat{\mathbf{x}}_{2,\ell+\eta_1-\eta_2}^l)$$
$$\bar{\mathbf{X}}_2^l = (\mathbf{x}_{2,1}^l, \mathbf{x}_{2,2}^l, ..., \mathbf{x}_{2,\eta_2}^l, \mathbf{x}_{2,\eta_2+1}^l, ...\mathbf{x}_{2,\ell}^l)$$

Under this setup, we can apply the Theorem A.11 of simplified case to $\bar{\mathbf{X}}_1^l$ and $\bar{\hat{\mathbf{X}}}_2^l$ to derive:

$$P(R_1 = \hat{R}_2) \geq \sum_{j=1}^V P_1(j)^2 -$$
$$\max_j \|\mathbf{w}_{a,j}\|_2 \cdot \|\mathbf{x}_{1,\ell+\eta_1-\eta_2}^L - \hat{\mathbf{x}}_{2,\ell+\eta_1-\eta_2}^L\|_2$$

where

$$\|\mathbf{x}_{1,\ell+\eta_1-\eta_2}^L - \hat{\mathbf{x}}_{2,\ell+\eta_1-\eta_2}^L\|_2 \propto \mathrm{MD}(\mathbf{X}_1^l, \hat{\mathbf{X}}_2^l)$$

and

$$\mathrm{MD}(\mathbf{X}_1^l, \mathbf{X}_2^l) = \mathrm{MD}(\mathbf{X}_1^l, \hat{\mathbf{X}}_2^l).$$

Now, we need to build correlation between $\hat{\mathbf{X}}_2^l$ and $\mathbf{X}_2^l$. We first omit the layer superscript $l$ to build a general formula. According to Lemma A.12, the difference between attention score $\mathbf{a}_2$ of $\mathbf{X}_2$ and the aggregated attention score $\hat{\mathbf{a}}_2'$ of $\hat{\mathbf{X}}_2$ is:

$$\|\mathbf{a}_2 - \hat{\mathbf{a}}_2'\|_2^2 = \sum_{i=1}^\ell a_{2,i}^2 \left(1 - \frac{\kappa_i}{\bar{\kappa}}\right)^2 \leq 2 = B_a.$$

According to the setup of $\hat{\mathbf{X}}_2$ and the definition of aggregated $\hat{\mathbf{a}}_2'$ in Lemma A.12, it is obvious that

$$(\hat{\mathbf{a}}_2')_{:\eta_2}(\mathbf{V}_2)_{:\eta_2} + (\hat{\mathbf{a}}_2')_{\eta_2+1:\ell}(\hat{\mathbf{V}}_2)_{\eta_1+1:\ell} = \hat{\mathbf{a}}_2\hat{\mathbf{V}}_2$$

where $\hat{\mathbf{V}}_2 = \mathrm{RMSNorm}(\hat{\mathbf{X}}_2)\mathbf{W}_V$. Then, suppose aggregated $\hat{\mathbf{X}}_2$ is $\hat{\mathbf{X}}_2' = \mathrm{concat}((\mathbf{X}_2)_{:\eta_2}, (\hat{\mathbf{X}}_2)_{\eta_1+1:\ell})$. In the same way as Theorem A.11, we apply Lemma A.8 to derive the difference between the attention output $\|\mathbf{h}_{2,\ell} - \hat{\mathbf{h}}_{2,\ell}'\|_2$:

$$\|\mathbf{h}_{2,\ell} - \hat{\mathbf{h}}_{2,\ell}'\|_2$$
$$\leq \varepsilon_v\sqrt{\ell}B_a + \sqrt{\sum_{j=1}^\ell (\frac{\|\mathbf{g}_1\|_\infty}{\sqrt{\epsilon_{RMS1}}}\|\mathbf{W}_V\|_2\|\mathbf{x}_{2,j} - \hat{\mathbf{x}}_{2,j}'\|_2)^2}$$
$$\leq 2\varepsilon_v\sqrt{\ell} + 2\frac{\|\mathbf{g}_1\|_\infty}{\sqrt{\epsilon_{RMS1}}}\|\mathbf{W}_V\|_2(\ell - \eta_2)\sqrt{d} \cdot \|\mathbf{g}_1\|_\infty$$
$$\leq 2\sqrt{d} \cdot \|\mathbf{g}_1\|_\infty\|\mathbf{W}_V\|_2(\sqrt{\ell} + \frac{\|\mathbf{g}_1\|_\infty}{\sqrt{\epsilon_{RMS1}}}(\ell - \eta_2))$$
$$= B_h$$

So, $B_h$ is a constant error for each layer. The bound of difference after the casual self-attention block is:

$$B_u = \|\mathbf{W}_O\|_2 B_h + \|\mathbf{x}_{1,\ell} - \mathbf{x}_{2,\ell}\|_2$$
$$\leq \|\mathbf{W}_O\|_2 B_h + 2\sqrt{d} \cdot \|\mathbf{g}_1\|_\infty$$

Since the $B_h$ is also an constant error, the final output after the FFN block $B_x$ is still a constant. Therefore, the norm of difference in inequality 4 is substituted with a constant $\hat{\varepsilon}$:

$$P(R_2 = \hat{R}_2)$$
$$\geq \sum_{j=1}^V P_2(j)^2 - \max_j \|\mathbf{w}_{a,j}\|_2 \cdot \hat{\varepsilon}$$

Then, the probability $P(R_1 = R_2)$ is

$$P(R_1 = R_2)$$
$$\geq \sum_{j=1}^V P_2(j)^2 + \sum_{j=1}^V P_1(j)^2 - \max_j \|\mathbf{w}_{a,j}\|_2 \cdot \hat{\varepsilon}$$
$$- \max_j \|\mathbf{w}_{a,j}\|_2 \cdot \|\mathbf{x}_{1,\ell+\eta_1-\eta_2}^L - \hat{\mathbf{x}}_{2,\ell+\eta_1-\eta_2}^L\|_2$$

Therefore,

$$P(R_1 = R_2) \propto \frac{1}{\mathrm{MD}(\mathbf{X}_1^l, \mathbf{X}_2^l)}$$

$\square$

# B. Proof of Theorem 5.5

We consider the problem of estimating the state value

$$V(s_t) = \mathbb{E}[R_t]$$

using terminal reward samples obtained from other states. Let

$$\mathcal{S} = \{s_1, \ldots, s_n\}$$

be a collection of sampled states, where each state $s_i$ is associated with a terminal reward $R_i$. For each $i$, define the similarity weight

$$P_{t,i} = \mathbb{P}(R_i = R_t).$$

We compare the following two estimators.

**Average estimator.**

$$\hat{V}_{\mathrm{avg}}(s_t) = \frac{1}{n} \sum_{i=1}^{n} R_i.$$

**Probability-weighted estimator.**

$$\hat{V}_{\mathrm{PW}}(s_t) = \frac{\sum_{i=1}^{n} P_{t,i} R_i}{\sum_{i=1}^{n} P_{t,i}}.$$

**Lemma B.1** (Bias Decomposition of the Probability-Weighted Estimator). *Let*

$$I \sim \mathrm{Unif}(\{1, \ldots, n\}),$$

*and define*

$$X = V(s_I) - V(s_t), \qquad W = P_{t,I}.$$

*Then*

$$\mathbb{E}[\hat{V}_{\mathrm{avg}}(s_t)] - V(s_t) = \mathbb{E}[X], \tag{5}$$

$$\mathbb{E}[\hat{V}_{\mathrm{PW}}(s_t)] - V(s_t) = \mathbb{E}[X] + \frac{\mathrm{Cov}(W, X)}{\mathbb{E}[W]}. \tag{6}$$

*Proof.* For the average estimator,

$$\mathbb{E}[\hat{V}_{\mathrm{avg}}(s_t)] = \frac{1}{n} \sum_{i=1}^{n} V(s_i)$$
$$= \mathbb{E}[V(s_I)]$$
$$= V(s_t) + \mathbb{E}[X].$$

This proves (5).

For the probability-weighted estimator,

$$\mathbb{E}[\hat{V}_{\mathrm{PW}}(s_t)] = \frac{\sum_{i=1}^{n} P_{t,i} V(s_i)}{\sum_{i=1}^{n} P_{t,i}}$$
$$= \frac{\mathbb{E}[W(V(s_t) + X)]}{\mathbb{E}[W]}$$
$$= V(s_t) + \frac{\mathbb{E}[WX]}{\mathbb{E}[W]}.$$

Using the covariance identity

$$\mathbb{E}[WX] = \mathbb{E}[W]\mathbb{E}[X] + \mathrm{Cov}(W, X),$$

we obtain

$$\mathbb{E}[\hat{V}_{\mathrm{PW}}(s_t)] - V(s_t) = \mathbb{E}[X] + \frac{\mathrm{Cov}(W, X)}{\mathbb{E}[W]}.$$

This proves 6. $\qquad\square$

**Assumption B.2** (Bias-Corrective Similarity Weighting). Let

$$I \sim \mathrm{Unif}(\{1, \ldots, n\}),$$

and define

$$X = V(s_I) - V(s_t), \qquad W = P_{t,I}.$$

Assume the similarity weights satisfy the following two conditions:

$$\mathbb{E}[X] \cdot \mathrm{Cov}(W, X) \le 0, \tag{7}$$

$$|\mathrm{Cov}(W, X)| \le 2\,\mathbb{E}[W]\,|\mathbb{E}[X]|. \tag{8}$$

**Theorem B.3** (Bias Improvement by Similarity Weighting). *Under Assumption B.2, the probability-weighted estimator has no larger absolute bias than the average estimator:*

$$\left|\mathbb{E}[\hat{V}_{\mathrm{PW}}(s_t)] - V(s_t)\right| \le \left|\mathbb{E}[\hat{V}_{\mathrm{avg}}(s_t)] - V(s_t)\right|.$$

*Proof.* From Lemma B.1,

$$\mathbb{E}[\hat{V}_{\mathrm{PW}}(s_t)] - V(s_t) = \mathbb{E}[X] + \frac{\mathrm{Cov}(W, X)}{\mathbb{E}[W]}.$$

Since $W = P_{t,I} \ge 0$, we have $\mathbb{E}[W] > 0$. Condition (7) implies that

$$\mathbb{E}[X] \quad \text{and} \quad \frac{\mathrm{Cov}(W, X)}{\mathbb{E}[W]}$$

always have opposite signs (or one of them is zero). Condition (8) further implies

$$\left|\frac{\mathrm{Cov}(W, X)}{\mathbb{E}[W]}\right| \le 2|\mathbb{E}[X]|.$$

We now prove

$$\left|\mathbb{E}[X] + \frac{\mathrm{Cov}(W, X)}{\mathbb{E}[W]}\right| \le |\mathbb{E}[X]|.$$

If $\mathbb{E}[X] \ge 0$, then

$$-2\mathbb{E}[X] \le \frac{\mathrm{Cov}(W, X)}{\mathbb{E}[W]} \le 0.$$

Therefore,

$$-\mathbb{E}[X] \leq \mathbb{E}[X] + \frac{\mathrm{Cov}(W,X)}{\mathbb{E}[W]} \leq \mathbb{E}[X],$$

which implies

$$\left| \mathbb{E}[X] + \frac{\mathrm{Cov}(W,X)}{\mathbb{E}[W]} \right| \leq |\mathbb{E}[X]|.$$

If $\mathbb{E}[X] \leq 0$, then

$$0 \leq \frac{\mathrm{Cov}(W,X)}{\mathbb{E}[W]} \leq -2\mathbb{E}[X].$$

Therefore,

$$\mathbb{E}[X] \leq \mathbb{E}[X] + \frac{\mathrm{Cov}(W,X)}{\mathbb{E}[W]} \leq -\mathbb{E}[X],$$

which again implies

$$\left| \mathbb{E}[X] + \frac{\mathrm{Cov}(W,X)}{\mathbb{E}[W]} \right| \leq |\mathbb{E}[X]|.$$

Finally, by Lemma B.1,

$$\mathbb{E}[\hat{V}_{\mathrm{avg}}(s_t)] - V(s_t) = \mathbb{E}[X].$$

Hence,

$$\left| \mathbb{E}[\hat{V}_{\mathrm{PW}}(s_t)] - V(s_t) \right| \leq \left| \mathbb{E}[\hat{V}_{\mathrm{avg}}(s_t)] - V(s_t) \right|.$$

$\square$

*Remark* B.4 (Interpretation of the Assumptions). Condition (7) characterizes the alignment between the similarity weights and the estimation error. Recall that the similarity weights are defined as

$$P_{t,i} = \mathbb{P}(R_i = R_t),$$

which measures the likelihood that state $s_i$ shares the same terminal outcome as the target state $s_t$. Intuitively, states whose values are closer to $V(s_t)$ are more likely to produce the same terminal reward and therefore tend to receive larger weights.

When the average estimator overestimates the target value, i.e.,

$$\mathbb{E}[X] > 0,$$

states with smaller value discrepancy

$$V(s_I) - V(s_t)$$

are expected to receive larger similarity weights, leading naturally to

$$\mathrm{Cov}(W,X) < 0.$$

Conversely, when the average estimator underestimates the target value, i.e.,

$$\mathbb{E}[X] < 0,$$

states with relatively larger values are expected to receive larger weights, yielding

$$\mathrm{Cov}(W,X) > 0.$$

Therefore, Condition (7) formalizes the intuition that the probability-weighted estimator tends to assign higher weights to states whose value estimates better align with the target state, thereby shifting the estimation bias toward zero.

Condition (8) controls the strength of this correction term. In practice, this can be enforced by introducing a temperature or scaling hyperparameter in the similarity weighting function, preventing the covariance term from over-correcting the original estimation bias.

## C. State Value Difference Distribution

The Figure 6 shows the distribution of the differences between the state values predicted by PPO and GRPO across all SVEB domains using Qwen2.5-1.5B as the frozen actor and trained critic. We observe a consistent phenomenon of state value degeneration, which provides additional empirical support for our observations.

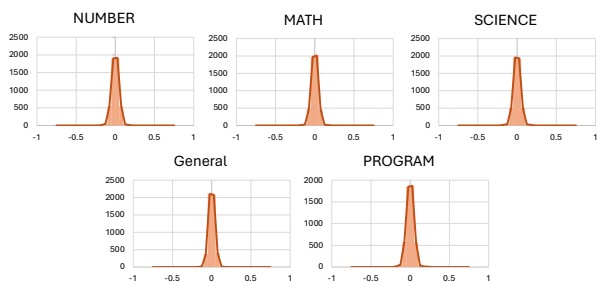

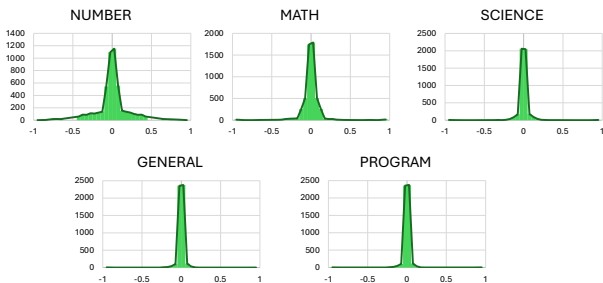

*Figure 6.* Distribution of the difference between state value predicted by PPO with $\lambda = 1.0$ and the group averaged reward. The x-axis is the difference and y-axis is the number of prediction.

The Figure 7 presents the distribution of differences between the state values predicted by *Numca* and GRPO across all SVEB domains under the same Qwen2.5-1.5B setup. In the SVEB-NUMBER domain, although the state values are clustered around the group-averaged reward, a non-negligible number of predictions fall outside this central region. Notably, we observe there is prediction near $0.5$ and $-0.5$, suggesting that *Numca* is able to identify "important" states that deterministically lead to correct or incorrect terminal outcomes. In contrast, for domains without explicit numerical milestones, *Numca* degenerates to behavior similar to GRPO, exhibiting a sharper distribution than PPO-N.

*Figure 7.* Distribution of the difference between state value predicted by *Numca* and the group averaged reward. The x-axis is the difference and y-axis is the number of prediction.

The Figure 8 illustrates the distribution of differences between the state values predicted by HISTA and GRPO across all SVEB domains using Qwen2.5-1.5B. Similar to NUMCA, the state values predicted by HISTA remain centered around the group-averaged reward. However, the resulting distributions are noticeably smoother, and predictions near $0.5$ and

$-0.5$ are consistently observed across domains. Moreover, HISTA maintains this smooth distribution across all SVEB tasks, demonstrating stronger generalization capability and corroborating the quantitative results reported in Table 4.

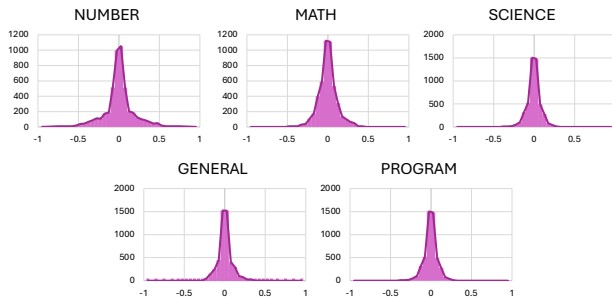

*Figure 8.* Distribution of the difference between state value predicted by *Hista* and the group averaged reward. The x-axis is the difference and y-axis is the number of prediction.

## D. SVEB on Different Models

Previous research has discovered the special shape and property of LLM or Transformer model hidden states (Godey et al., 2024; Rudman et al., 2023), which raise the concern about richness or quality of hidden states as an representation of state. We consider LLMs following large-scale pretraining should be capable for producing meaning representation. To support this, we provide SVEB results across the Qwen2.5 and Qwen3 series in Table 9. Throughout different setups, for example different model series and whether the model is instruction tuned, *Hista* achieve consistent improvement compared with GRPO. For small models like Qwen2.5-0.5B and Qwen3-0.6B, the result proves that the quality of their hidden states is enough to serve as representation.

## E. Ablation Study

In this section, we perform ablation study on hyperparameter of *Hista*. For efficiency, we use Qwen2.5-1.5B-Instruct as the actor and SVEB-NUMBER as the benchmark for ablation study. First component is the *MinDistance*, which is the foundation of *Hista*. We evaluate two different designs:

1. Substitute the L2 distance with $(1 - \text{Cosine Similarity})$

2. Change the *MinDistance* operation to sum of distance between hidden states in the same position, which is $\sum_{i=1}^{n} \| \mathbf{x}_{1,i} - \mathbf{x}_{2,i} \|$.

The result is presented in Table 10. Substituting the *MinDistance* causes significant performance drop, which validates the Theorem 5.2 empirically.

*Table 9.* Mean Absolute Error (MAE) for GRPO and *Hista* across SVEB fields and different models, with lower values being better. All values are calculated relative to a reference value established by MCS@20.

| SVEB | Number ↓ | Math ↓ | Science ↓ | General ↓ | Programming ↓ |
|---|---|---|---|---|---|
| Qwen2.5-0.5B-Instuct | | | | | |
| GRPO@40 | 0.134 | 0.153 | 0.195 | 0.120 | 0.160 |
| HISTA@40 | $0.108_{\downarrow -0.026}$ | $0.143_{\downarrow -0.010}$ | $0.171_{\downarrow -0.024}$ | $0.094_{\downarrow -0.026}$ | $0.134_{\downarrow -0.026}$ |
| MCS@1 | 0.150 | 0.197 | 0.226 | 0.096 | 0.141 |
| MCS@2 | 0.107 | 0.151 | 0.168 | 0.076 | 0.098 |
| Qwen2.5-1.5B-Instuct | | | | | |
| GRPO@40 | 0.178 | 0.161 | 0.195 | 0.176 | 0.154 |
| HISTA@40 | $0.132_{\downarrow -0.046}$ | $0.139_{\downarrow -0.022}$ | $0.166_{\downarrow -0.029}$ | $0.132_{\downarrow -0.044}$ | $0.118_{\downarrow -0.036}$ |
| MCS@1 | 0.194 | 0.236 | 0.225 | 0.155 | 0.176 |
| MCS@2 | 0.146 | 0.156 | 0.168 | 0.124 | 0.120 |
| Qwen2.5-7B-Instuct | | | | | |
| GRPO@40 | 0.212 | 0.176 | 0.196 | 0.162 | 0.142 |
| HISTA@40 | $0.164_{\downarrow -0.048}$ | $0.140_{\downarrow -0.036}$ | $0.152_{\downarrow -0.044}$ | $0.148_{\downarrow -0.014}$ | $0.112_{\downarrow -0.030}$ |
| MCS@1 | 0.173 | 0.182 | 0.199 | 0.226 | 0.141 |
| MCS@2 | 0.114 | 0.131 | 0.146 | 0.152 | 0.101 |
| Qwen3-0.6B-Instuct, Disable Thinking | | | | | |
| GRPO@40 | 0.168 | 0.155 | 0.124 | 0.147 | 0.212 |
| HISTA@40 | $0.159_{\downarrow -0.009}$ | $0.151_{\downarrow -0.004}$ | $0.115_{\downarrow -0.009}$ | $0.120_{\downarrow -0.027}$ | $0.184_{\downarrow -0.028}$ |
| MCS@1 | 0.267 | 0.281 | 0.186 | 0.186 | 0.208 |
| MCS@2 | 0.180 | 0.193 | 0.130 | 0.129 | 0.142 |
| Qwen3-1.7B-Instuct, Disable Thinking | | | | | |
| GRPO@40 | 0.175 | 0.136 | 0.125 | 0.137 | 0.154 |
| HISTA@40 | $0.141_{\downarrow -0.034}$ | $0.122_{\downarrow -0.014}$ | $0.110_{\downarrow -0.015}$ | $0.105_{\downarrow -0.032}$ | $0.128_{\downarrow -0.026}$ |
| MCS@1 | 0.212 | 0.208 | 0.142 | 0.152 | 0.160 |
| MCS@2 | 0.152 | 0.160 | 0.117 | 0.106 | 0.116 |
| Qwen3-4B-Instuct, Disable Thinking | | | | | |
| GRPO@40 | 0.178 | 0.156 | 0.125 | 0.147 | 0.178 |
| HISTA@40 | $0.122_{\downarrow -0.056}$ | $0.139_{\downarrow -0.017}$ | $0.102_{\downarrow -0.023}$ | $0.118_{\downarrow -0.029}$ | $0.165_{\downarrow -0.013}$ |
| MCS@1 | 0.209 | 0.226 | 0.145 | 0.159 | 0.199 |
| MCS@2 | 0.139 | 0.149 | 0.101 | 0.106 | 0.146 |
| Qwen3-8B-Instuct, Disable Thinking | | | | | |
| GRPO@40 | 0.183 | 0.146 | 0.156 | 0.141 | 0.182 |
| HISTA@40 | $0.127_{\downarrow -0.056}$ | $0.125_{\downarrow -0.021}$ | $0.129_{\downarrow -0.027}$ | $0.120_{\downarrow -0.021}$ | $0.180_{\downarrow -0.002}$ |
| MCS@1 | 0.214 | 0.219 | 0.206 | 0.208 | 0.204 |
| MCS@2 | 0.145 | 0.147 | 0.131 | 0.126 | 0.156 |
| Qwen3-4B-Base | | | | | |
| GRPO@40 | 0.223 | 0.223 | 0.166 | 0.147 | 0.108 |
| HISTA@40 | $0.146_{\downarrow -0.077}$ | $0.139_{\downarrow -0.084}$ | $0.144_{\downarrow -0.022}$ | $0.107_{\downarrow -0.040}$ | $0.078_{\downarrow -0.030}$ |
| MCS@1 | 0.196 | 0.190 | 0.107 | 0.136 | 0.052 |
| MCS@2 | 0.139 | 0.140 | 0.084 | 0.080 | 0.058 |
| Qwen3-8B-Base | | | | | |
| GRPO@40 | 0.144 | 0.173 | 0.226 | 0.164 | 0.135 |
| HISTA@40 | $0.093_{\downarrow -0.051}$ | $0.135_{\downarrow -0.038}$ | $0.182_{\downarrow -0.044}$ | $0.103_{\downarrow -0.061}$ | $0.089_{\downarrow -0.046}$ |
| MCS@1 | 0.062 | 0.165 | 0.225 | 0.083 | 0.094 |
| MCS@2 | 0.056 | 0.131 | 0.153 | 0.055 | 0.068 |

*Table 10.* Ablation study on the distance measurement of *Hista*. "Cosine" means the distance is measured by $(1 - \text{Cosine Distance})$. "Corresponding" means the distance is measure by $\sum_{i=1}^{n} \| \mathbf{x}_{1,i} - \mathbf{x}_{2,i} \|$.

| METHOD | MINDISTANCE | COSINE | CORRESPOND |
|---|---|---|---|
| SVEB | **0.142** | 0.226 | 0.208 |

*Table 11.* Ablation study on the index of layer that we take the hidden state from.

| Value | 1 | 2 | 3 | 6 | 10 |
|---|---|---|---|---|---|
| SVEB | 0.142 | 0.144 | 0.141 | 0.146 | 0.142 |

In the practical implementation of *Hista*, another hyperparameter is the index $l$ of layer that we take the hidden state from. Although Theorem 5.2 shows the correlation exists in any layer, we try to find whether the layer in the middle will bring better performance (Skean et al., 2025). The result is presented in Table 11. Changing the index of layer does not bring notable performance change. For simplicity, we always use the last hidden layer in all other experiments and training.

Since the EMA and pooling operation in the implementation of *Hista* introduces factor $\alpha$ and compression interval $\varphi$, we investigate their influence on state value estimation in Table 12. The result shows that settings $\alpha = 0$ and $\varphi = 1$ can achieve best performance in state value estimation. For other value of $p$, the best performance is achieved only with a special $\alpha$. One concern is that $p = 1$ will introduce more computation cost, and there is not significant benefit according to SVEB. Therefore, we always use the combination fo $\varphi = 5$ and $\alpha = 0.7$.

Finally, the last two hyperparameter in the state sampling interval $\delta$ and $k$ nearest neighbor. We fix the $k = 40$ and try different $\delta$ value in Table 13. The result also indicates that we also need to design the combination fo $\delta$ and $k$ value to get the optimal performance. In our experiments, we use a heuristic combination $k = 66$ and $\delta = 50$.

## F. Efficiency of Hista

To prove the claim that *Hista*'s latency and GPU memory footprint is comparable with GRPO, we monitor the aver-

*Table 12.* Ablation study on the SVEB metric with different EMA factor $\alpha$ (x-axis) and the compression interval $\varphi$ y-axis.

| Value | 0.0 | 0.5 | 0.6 | 0.7 | 0.8 | 0.9 |
|---|---|---|---|---|---|---|
| 1 | **0.140** | 0.142 | 0.144 | 0.146 | 0.151 | 0.157 |
| 5 | 0.157 | 0.146 | 0.143 | **0.142** | 0.142 | 0.145 |
| 10 | 0.155 | 0.146 | 0.144 | **0.143** | 0.144 | 0.145 |
| 25 | 0.164 | 0.157 | 0.158 | 0.154 | **0.151** | 0.154 |

*Table 13.* Ablation study on the sampling interval value with fixed $k = 66$.

| Value | 10 | 25 | 50 | 100 | 150 |
|---|---|---|---|---|---|
| SVEB | 0.146 | 0.144 | **0.142** | 0.142 | 0.148 |

*Table 14.* Average step time of the first 1200 training step

| Time (s) | Qwen2.5-14B | Qwen3-4B | Qwen3-1.7B |
|---|---|---|---|
| DAPO | 20.54 | 62.18 | 24.81 |
| +Hista | 21.07 | 62.82 | 25.01 |

age step time and average GPU memory consumption of the first 1200 training step of different models (one step refers to one policy update) in Table 14 and Table 15. All models are Base version. The reponse length of Qwen2.5-14B and Qwen3-1.7B is set to 4096. The reponse length of Qwen3-4B is set to 8192. In some settings. *Hista* uses less memory, because our implementation explicitly calls torch.cuda.empty_cache() to reduce fragmentation after the computation of *Hista*. Given the *Hista* algorithm has additional torch.cuda.empty_cache() call, the average additional latency introduced by *Hista* is $< 1$ second per training step across all configurations. It is because, according to the pipeline in Figure 4, the stage 1 is completed together with the compute_old_logp() function, the stage 2 only contains redistribution of hidden states, which has negligible cost, and the computational cost of top_k in stage 4 is also low. The only heavy computation happens in stage 3, but it can be accelerated by collecting the embeddings into one matrix on GPU and applying torch.cummin() and torch.cumsum().

## G. Training Details

### G.1. Training of Numca

For Qwen2.5-1.5B-Instruct model, the data is sampled from DAPO-17K and OpenR1-220K dataset. The data is further filtered by $0.1 < \text{accuracy} < 0.8$, where the accuracy is calculated through sampling 20 responses using Qwen2.5B-1.5B-Instruct. It results in train set with 8199 problems and validation set with 910 problems, where approximately 1/3 from DAPO-17K and 2/3 from OpenR1-220K dataset. We implement the *Numca* algorithm on the TRL library to perform RL training on 4xRTX3090 GPU.

*Table 15.* Average GPU memory consumption of the first 1200 training step.

| Mem (GB) | Qwen2.5-14B | Qwen3-4B | Qwen3-1.7B |
|---|---|---|---|
| DAPO | 1011 | 582 | 189 |
| +Hista | 1020 | 486 | 182 |

*Table 16.* From top to bottom, the first part is statistics of train set of 1.5B model; The second part is statistics of 3B, 7B, and 14B model.

| Source | Count |
|---|---|
| DAPO-17K | 2500 |
| OpenR1-220K | 3000 |
| Llama-Nemotron Post Training | 3000 |
| WebInstruct-verified | 3000 |
| Verifiable Coding Problem | 2291 |
| DAPO-17K | 2500 |
| OpenR1-220K | 2500 |
| Llama-Nemotron Post Training | 3000 |
| WebInstruct-verified | 3000 |
| Verifiable Coding Problem | 2400 |

For Qwen2.5-Math-1.5B-Instruct model, we adopt the same pipeline as the Qwen2.5-1.5B-Instruct model to construct the train set and directly use the validation set of Qwen2.5-1.5B-Instruct model. It results in 12334 problems in train set with similar distribution. We use the same code and 4xRTX3090 GPU to train.

### G.2. Training of Hista

*Hista* is also implemented with TRL library. For model with different number of parameters, we will use different settings to get desirable training speed.

**1.5B and 0.5B model** We use 4xRTX3090 GPU and for DeepSpeed Zero 3 (Rajbhandari et al., 2020).

**3B model** We use 8xH800 GPU with DeepSpeed Zero 2.

**7B and 14B model** We use 2 nodes equipped with 8xH800 GPU and DeepSpeed Zero 3.

**R1-Distill-1.5B** We use 4 nodes equipped with 8xH800 GPU and DeepSpeed Zero 2.

The distribution of training data is presented in Appendix H.

## H. Data Statistics

For MATH and DAPO-17K dataset, we directly use the original dataset without any filter or processing. For **Hybrid Reasoning Dataset**, each model has its own train set. To construct the train set for different model, we will generate 20 rollouts from the corresponding model and filter questions with accuracy $< 0.1$ or accuracy $> 0.8$. Then, the training set for this model is sampled from remaining questions The statistic for 1.5B, 3B, 7B, and 14B model in presented in Table 16.

## I. More Discussions

**Performance of *Hista*:** We provide the following analysis to understand regimes where Hista may underperform compared to group-average baselines such as DAPO in some benchmarks of Table 5. Consider four disjoint trajectories:

$$s_0 \rightarrow s_{1,1} \rightarrow s_{1,2} \rightarrow s_{1,3} \rightarrow s_{1,\text{correct}},$$

$$s_0 \rightarrow s_{2,1} \rightarrow s_{2,2} \rightarrow s_{2,3} \rightarrow s_{2,\text{correct}},$$

$$s_0 \rightarrow s_{3,1} \rightarrow s_{3,2} \rightarrow s_{3,3} \rightarrow s_{3,\text{wrong}},$$

$$s_0 \rightarrow s_{4,1} \rightarrow s_{4,2} \rightarrow s_{4,3} \rightarrow s_{4,\text{wrong}}.$$

Assume that states from different trajectories are well separated in the representation space. Then, the value estimate of an intermediate state, e.g., $V(s_{1,1})$, is mainly determined by continuations passing through that state. Since all continuations from $s_{1,1}$ are successful, *Hista* assigns it a high value, concentrating advantage on transitions entering such states (e.g., $s_0 \rightarrow s_{1,1}$) while assigning weaker signals to later transitions. Therefore, *Hista* tends to emphasize the **observed key reasoning steps** instead of uniformly reinforcing the entire successful trajectory. In contrast, group-average methods such as DAPO assign identical advantages to all steps in a successful trajectory. While this is less precise, it ensures full credit coverage, meaning every step that contributes to success is reinforced. Consequently, Hista is more effective when accurate value estimation and gradient denoising are critical, whereas group-average methods may remain competitive when uniform reinforcement over long trajectories is more beneficial.

*Table 17.* Notation Reference Table for SVEB, Numca, Hista and the Proof Section

| Notation | Meaning | Domain/Note |
|---|---|---|
| **State Value Estimation Benchmark** | | |
| $\mathcal{D}_s$ | Dataset of sampled states $\{s_t^{(j)}\}_{j=1}^N$ | Set |
| $s_t^{(j)}$ | The $j$-th sampled state at step $t$ | $s_t^{(j)} \in \mathcal{D}_s$ |
| $N$ | Total number of states in the dataset $\mathcal{D}_s$ | $N = |D_s|$ |
| $n$ | Number of independent MCS per state | $n = 20$ for reference value |
| $r(s_T^{(i)})$ | Terminal reward of the $i$-th Monte Carlo continuation | Section 3.1 |
| $\widehat{V}(s_t)$ | Monte Carlo reference value (average of terminal rewards) | Equation 4.1 |
| $f(s_t, \theta)$ | Value estimation method/function being evaluated | $\theta$ includes parameters and rollouts |
| $\widehat{V}_{f,\theta}(s_t)$ | Estimated state value produced by method $f$ | $\widehat{V}_{f,\theta}(s_t) = f(s_t, \theta)$ |
| $\mathrm{MAE}(f, D_s)$ | Mean Absolute Error as the metric for method $f$ on $D_s$ | Equation 4.1 |
| **Numerical Milestone Credit Assignment** | | |
| $\mathcal{P}$ | Predefined set of numerical patterns | Set of Strings |
| $m$ | A milestone; a token subsequence matching a pattern $p \in \mathcal{P}$ | $\mathrm{decode}(m) = p$ |
| $\mathbb{M}$ | Function mapping a state to the set of all milestones contained within it | $s_t^M \triangleq \mathbb{M}(s_t)$ |
| $s_t^M$ | Abstract state at time $t$, defined as the set of milestones $m$ | Concept from HRL |
| $a_t^M$ | Macro action; the sequence of tokens generated between two abstract states | Concept from HRL |
| $\mathcal{D}_r$ | Set of rollouts from same input problem | A group of rollouts |
| $\tau$ | A single rollout $(s_0, a_0, \ldots, s_T)$ | Section 3.1 |
| $\mathcal{T}$ | Dictionary/Table storing counts and reward sums for each abstract state | $\mathcal{T}[s^M] = (\mathrm{count}, \mathrm{reward\_sum})$ |
| $V(s^M)$ | Estimated value of an abstract state, computed via reward averaging | Algorithm 1 |
| **Hidden State based State Value Estimation** | | |
| $\mathbf{x}$ | One hidden state tensor | $\mathbf{x} \in \mathbb{R}^d$ |
| $\mathbf{X}$ | Sequence of hidden states $(\mathbf{x}_1, \ldots, \mathbf{x}_\eta)$ | $\mathbf{X} \in \mathbb{R}^{\eta \times d}$ |
| $\eta$ | Number of tokens/hidden states in a sequence | $\eta = |s_t|$ |
| $\mathrm{MD}(\mathbf{X}_1, \mathbf{X}_2)$ | *MinDistance* operation between two hidden state sequences | Definition 5.1 |
| $R_i$ | Terminal reward of the continuation from state $s_i$ | Random Variable |
| $P(R_1 = R_2)$ | Probability that continuation of two states share the same terminal reward | $P(R_1 = R_2) \in [0, 1]$ |
| $L$ | Total number of layers in the LLM | LLM architecture |
| $\mathcal{N}$ | Number of states in one response group | $\mathcal{N} \in \mathbb{N}$ |
| $\widehat{V}_{\mathrm{avg}}$ | Naive average estimator of the state value | Definition 5.3 |
| $\widehat{V}_{\mathrm{PW}}$ | Probability-weighted estimator of the state value | Definition 5.4 |
| $P_{t,i}$ | Weight (probability) between state $s_t$ and state $s_i$ | $P_{t,i} = P(R_t = R_i)$ |
| $\alpha$ | Smoothing factor for Exponential Moving Average | $\alpha \in [0, 1]$ |
| $\varphi$ | Compression interval for compressing hidden states | $\varphi \in \mathbb{N}, \varphi \le \delta$ |
| $\mathbf{E}_\tau$ | Sequence of compressed embeddings for rollout $\tau$ | $\mathbf{E}_\tau \in \mathbb{R}^{\lfloor \eta/\varphi \rfloor \times d}$ |
| $\delta$ | Sampling interval for defining the finite state space $\mathcal{S}_\tau$ | $\delta \in \mathbb{N}, \delta \ge \varphi$ |
| $s_{\tau,i}^H$ | Abstract state at index $i$, represented by embeddings | $s_{\tau,i}^H \in \mathbb{R}^{(i\delta) \times d}$ |
| $k$ | Number of nearest neighbors used for value estimation | $k \in \mathbb{N}$ |
| $\omega_i$ | Probability-based weight assigned to the $i$-th neighbor | Equation 5.2 |
| **Proof** | | |
| $\mathbf{G}$ | Diagonal matrix with elements of $\mathbf{g}$ on the main diagonal | $\mathbf{G} \in \mathbb{R}^{d \times d}$ |
| $y(\mathbf{x})$ | Unscaled normalized mapping of $\mathbf{x}$ | $y(\mathbf{x}) \in \mathbb{R}^d$ |
| $J_y(\mathbf{x})$ | Jacobian matrix of the unscaled mapping $y(\mathbf{x})$ | $J_y(\mathbf{x}) \in \mathbb{R}^{d \times d}$ |
| $\varepsilon_{qk}$ | Maximum norm among the query and key vectors | $\varepsilon_{qk} \in \mathbb{R}^+$ |
| $\varepsilon_v$ | Uniform upper bound on the $\ell_2$-norm of value rows | $\varepsilon_v \in \mathbb{R}^+$ |
| $M_x, M_h$ | Global upper bound of input $\mathbf{x}_i$ and intermediate $\mathbf{h}_i$ | $M_x, M_h \in \mathbb{R}^+$ |
| $\Omega$ | Compact (bounded and closed) domain of intermediate states | $\Omega \subset \mathbb{R}^d$ |
| $L_{\mathrm{SwiGLU}}$ | Local Lipschitz constant of SwiGLU over domain $\Omega$ | $L_{\mathrm{SwiGLU}} \in \mathbb{R}^+$ |
| $d_{TV}(P_1, P_2)$ | Total Variation (TV) distance between distributions $P_1$ and $P_2$ | $d_{TV} \in [0, 1]$ |
| $\mathrm{range}(P_1)$ | The difference between the maximum and minimum probabilities in $P_1$ | $\mathrm{range}(P_1) \in [0, 1]$ |
| $\ell$ | Length of elongated hidden states up to the token index of action output | $\ell \in \mathbb{N}^+, \ell > \eta$ |
| $\epsilon_{RMS1}^l, \epsilon_{RMS2}^l$ | Regularization constants for RMSNorm inside layer $l$ | $\epsilon \in \mathbb{R}^+$ |
| $B$ | Progressive sensitivity upper bounds for each component's outputs | $B \in \mathbb{R}^+$ |
| $C$ | Layer-dependent analytical contraction/expansion multiplier constants | $C \in \mathbb{R}^+$ |
| $\hat{r}_i$ | Replication count for the $i$-th coordinate in the first block | $\hat{r}_i \in \mathbb{Z}_{\ge 0}, \sum_{i=1}^{\eta_2} \hat{r}_i = \eta_1$ |
| $\kappa_i$ | Scaling coefficient acting as a surrogate weight for index $i$ | $\kappa_i \ge 0$ |

