# OpenReview forum: "Hista and Numca: Estimate State Value Effectively for Large Language Model Reinforcement Learning"
_ICML.cc/2026/Conference — ICML 2026 spotlight_

### Official Review · Reviewer_if7k · 2026-03-12

**Soundness:** 3
**Presentation:** 2
**Significance:** 2
**Originality:** 2
**Overall Recommendation:** 4
**Confidence:** 2

**Summary:**

This paper investigates the problem of state value estimation in LLM reinforcement learning. To explore this issue, the authors introduce the State Value Estimation Benchmark (SVEB). Through empirical study based on this benchmark, they find that the predicted state values of the standard PPO’s critic in LLM collapse towards the group average reward. Based on this insight, the paper proposes Numca, a method that leverages numerical patterns as structured milestones to guide credit assignment in mathematical problem-solving, and Hista, a hidden state-based nearest neighbor estimator that utilizes similarity-weighted averages of terminal rewards. Improved value estimation on SVEB reduces Mean Absolute Error (MAE) and, in downstream RL training tasks, sometimes enhances the training performance of DAPO and CSIPO variants.

**Compliance With Llm Reviewing Policy:**

Affirmed.

**Key Questions For Authors:**

1. Please explain Weakness 1 and Weakness 2 in detail. In particular, the explanation of Weakness 1 will significantly influence the assessment of a positive attitude toward this paper.

2. Considering the fluctuations and relatively small differences shown in the reward curves in Figure 1, how were the final checkpoints selected for the evaluations reported in Tables 3, 5, 6, and 7? Are these results derived from a single run's peak performance (which carries the risk of cherry-picking), or are they averaged results from multiple random seeds? Please provide the standard deviations for the results.

3. The claim regarding the efficiency of Hista lacks support from experimental results. On page 6, the paper states that Hista is "as efficient as GRPO," yet the method illustrated in Figure 4 involves hidden state extraction, pooling, immediate state collection, pairwise distance computation, kNN retrieval, and weighted averaging. Please provide a specific table comparing the training time and GPU memory footprint of the standard algorithm and its Hista variant to demonstrate the additional overhead introduced by Hista.

4. In Section 5.1, the paper mentions, "Numca estimates the value of an abstract state $s ^ M$ by averaging the final rewards of all rollouts that contain $s ^ M$; this value is then assigned uniformly to every token inside the corresponding macro action." Please explain the reason for doing so. From a classical RL perspective, the action-value function $Q(s ^ M _ t, a ^ M _ t)$ corresponding to the state $s ^ M _ t$ and action $a ^ M _ t$ would differ between actions. Therefore, it seems questionable to simply assign the state value $V(s ^ M _ t)$ of the state $s ^ M _ t$ uniformly to every token within the corresponding macro action.

**Limitations:**

This paper does not discuss the limitations of the proposed method. For example, this paper doesn’t analyze the scope of the assumptions used in the theoretical framework presented, as well as the additional computational cost incurred by the proposed approach.

**Strengths And Weaknesses:**

Strength
1. For LLM reinforcement learning, token-level value estimation is an important yet underexplored problem.

2. Constructing SVEB facilitates quantitative analysis of state value degradation, which is beneficial for elucidating the problem. Furthermore, employing Monte Carlo methods to generate intermediate states and estimate reference values provides a reasonable approach to concretize the state value estimation problem.

3. Based on the investigated problem, this paper proposes Numca, which focuses on mathematical reasoning tasks, and Hista, a more general method. Empirical results show that Numca reduces state value estimation errors on mathematical tasks, while Hista effectively lowers value estimation errors across a broader range of tasks.

Weakness
1. There appears to be an inconsistency between the theory and the proposed Hista method. Theorem 5.2 states that the probability of two states receiving the same reward is proportional to MinDistance, and the Similarity-Weighted estimator uses this probability to weight the rewards. However, $w_j$ is defined using the reciprocal of MinDistance, establishing an inverse relationship, which directly contradicts Theorem 5.2. This inconsistency requires detailed explanation, as it significantly impacts the assessment of the proposed method.

2. The paper may be overstating its claims regarding consistent and stable accuracy gains, and a more nuanced analysis of the experimental results is required. The paper repeatedly states that Hista consistently improves performance, yet the results presented in Tables 5 and 6 appear more complex. For instance, such as "MATH" in Table 5, aside from the performance improvement on the 0.5B DAPO variant and the marginal gain on the 14B DAPO variant, all other variants exhibit varying degrees of performance degradation.

3. This paper suffers from omissions and ambiguities in its presentation, with even the use of symbols causing confusion, hindering readers' comprehension of the proposed methods. For instance, the critic loss in Section 3.2 appears to lack a gradient stopping term, the variable $n$ is used inconsistently across different sections (e.g., in Definition 5.1 and the Similarity-Weighted estimator) and so on.

---

> ### Author Rebuttal · Authors · 2026-03-31
>
> > ## Overall Response
>
> We thank the reviewer for the careful reading and helpful feedback. In this rebuttal, we clarify the main concerns regarding the theoretical formulation, experimental claims, and presentation. We will revise the paper to improve clarity, correct inconsistencies, and better align our claims with the empirical results. We also provide additional details on evaluation and efficiency to support our conclusions.
>
> > ## Weakness 1: Inconsistency in Theorem 5.2
>
> We apologize for this oversight. The issue stems from unclear use of “\propto”. The intended relationship is that the probability of two states receiving the same reward is proportional to the inverse of MinDistance. We will revise Theorem 5.2 to explicitly state:
> $$
> P(R_1 = R_2) \propto \frac{1}{\mathrm{MD}(\mathbf{X}^l_1, \mathbf{X}^l_2)}, \quad \forall l\in[1, L]
> $$
>
> > ## Weakness 2: Consistent gain claim
>
> For further details regarding experimental settings and an analysis of the limited performance gains, please refer to Reviewer Chui's "Key Question 1" section and Reviewer f7ps's "Weakness 4" section. Our goal is to show that Hista achieves stronger *average* performance across models/algorithms. While gains vary, results support the core idea: improving value estimation via hidden-state grouping.
>
> > ## Weakness 3: Use of Symbols
>
> We acknowledge the ambiguity caused by inconsistent notation. We will (1) clarify symbols within each section and (2) add a symbol table in the appendix.
>
> Specifically, in the Hista section, we used $n$ for both trajectory length and number of states. We will replace the latter with $N$ to avoid confusion.
>
> > ## Key Question 1: Presentation issues
>
> See “Weakness 1” and “Weakness 3”. We will ensure all symbols are clearly defined and consistent, and fix Theorem 5.2.
>
> > ## Key Question 2: Checkpint Selection
>
> **Checkpoint selection:** We select the best checkpoint using a held-out validation set, then evaluate on test benchmarks, following standard practice.
>
> **Evaluation settings:** Most benchmarks use temperature = 0.0; for harder tasks (AIME, AMC), we use temperature = 0.1 and report avg@16.
>
> **Single run vs. multiple seeds:** Due to the high cost of LLM training, we report single-run results. However:
> 1. Selection is based only on validation performance.
> 2. Training curves (Fig. 1) show stable convergence, suggesting limited variance.
>
> **Statistical validation:** To strengthen reliability, we report:
> - 95% confidence intervals
> - t-test p-values
> - probability of superiority
>
> | | 95%CI | p-value (p < 0.05 means significant) | Probability of Superiority |
> | ---- | ---- | ---- | ---- |
> | 0.5B DAPO | [0.184, 0.207] | - | - |
> | 0.5B DAPO+Hista | [0.200, 0.224] | 0.005 | 97.68 |
> | 1.5B DAPO | [0.355, 0.390] | - | - |
> | 1.5B DAPO+Hista | [0.363, 0.398] | 0.454 | 69.76 |
> | 1.5B CSIPO | [0.360, 0.397] | - | - |
> | 1.5B CSIPO+Hista | [0.369, 0.406] | 0.225 | 75.55 |
> | 3B DAPO | [0.433, 0.469] | - | - |
> | 3B DAPO+Hista | [0.438, 0.475] | 0.467 | 65.12 |
> | 3B CSIPO | [0.429, 0.460] | - | - |
> | 3B CSIPO+Hista | [0.436, 0.466] | 0.351 | 68.27 |
> | 7B DAPO | [0.517, 0.545] | - | - |
> | 7B DAPO+Hista | [0.523, 0.552] | 0.512 | 63.19 |
> | 7B CSIPO | [0.508, 0.538] | - | - |
> | 7B CSIPO+Hista | [0.515, 0.546] | 0.334 | 68.62 |
> | 14B DAPO | [0.549, 0.595] | - | - |
> | 14B DAPO+Hista | [0.561, 0.645] | 0.257 | 73.61 |
>
> > ## Key Question 3: Efficiency of Hista
>
> **GPU Memory Footprint**
>
> We monitor average GPU memory consumption over the first 1200 steps:
>
> | Average GPU Mem (GB) | Qwen3-8B-Base (8096 context) | Qwen3-4B-Base (8096 context) | Qwen3-1.7B-Base (3072 context) | Qwen2.5-14B (4096 context) |
> | ---- | ---- | ---- | ---- | ---- |
> | DAPO | 998 | 582 | 189 | 1011 |
> | DAPO + Hista | 948 | 486 | 182 | 1020 |
>
> In some settings Hista uses less memory due to explicit `torch.cuda.empty_cache()` reducing fragmentation.
>
> **Training Time Overhead**
>
> As detailed in our response to Reviewer f7ps ("Hista Training Efficiency" section), given the Hista algorithm has additional `torch.cuda.empty_cache()` call, the average additional latency introduced by Hista is **<1 second per training step** across all configurations. Therefore, Hista introduces nearly no overhead, but brings improvement.
>
> > ## Key Question 4
>
> We agree token-level Q-values may differ. However, uniform assignment is a deliberate design choice:
> 1. **Granularity Constraint:** Numca operates based on **numerical milestones**. A macro action is defined as the sequence of tokens between two state abstraction. By definition, there are no intermediate milestones within a macro action to provide finer-grained value signals. Consequently, we lack the structural information required to decompose the macro action further.
> 2. **Robustness:** Given the absence of intermediate rewards or milestones, this uniform assignment is the most robust practice. It avoids introducing arbitrary variance that might arise from attempting to interpolate values without supporting signals.

---

> > ### Author Rebuttal · Reviewer_if7k · 2026-04-06
> >
> > The authors have addressed most of my concerns. I would like to keep my original positive rating score.

---

### Official Review · Reviewer_f7PS · 2026-03-13

**Soundness:** 4
**Presentation:** 4
**Significance:** 3
**Originality:** 3
**Overall Recommendation:** 5
**Confidence:** 3

**Summary:**

This paper studies state value estimation in RL for LLMs and analyzes the limitations of critic models used in standard RL algorithms. The authors show that existing critics often become uninformative because their value predictions collapse toward the group-averaged reward. To address this issue, the paper reframes value estimation as a sparse-reward goal-reaching problem and proposes two methods:
- Numca uses numerical values appearing in reasoning traces as milestones to estimate the value of intermediate tokens.
- Hista represents each state using the set of hidden embeddings of tokens in the response and estimates state values via a weighted sum of rewards from nearest neighboring states.
Empirical results show that both Numca and Hista provide more accurate value estimation and help stabilize and improve the training of online RL algorithms such as GRPO, DAPO, and CSIPO.

**Compliance With Llm Reviewing Policy:**

Affirmed.

**Final Justification:**

The paper makes a strong and compelling contribution to the problem of value estimation in RL training for LLMs. The research question is relevant, and the proposed methods are both intuitive and technically well-supported by the experiments.
The paper is well written, with clear motivation, solid mathematical formulation, and explanations throughout.
I recommend accepting the paper.

**Key Questions For Authors:**

1. Is there a reason why the authors chose Qwen2.5 as the primary experimental model rather than Qwen3, given that the latter has been available for some time?
2. Could the authors provide training dynamics for Hista experiments as well? Since such curves are presented for Numca, it would be helpful to include them for Hista for consistency.
3. How are Numca and Hista integrated into GRPO and DAPO? Do the authors still generate groups of samples per prompt? The purpose of multi-sampling in GRPO is to estimate value signals in the absence of a critic model. With Numca and Hista providing explicit value estimates, it is unclear whether multi-sampling is still necessary.
4. Is there a reason why the authors did not experiment with combining Numca/Hista with PPO?

**Strengths And Weaknesses:**

# Strengths
- The research problem, improving value estimation for RL training of LLMs, is both important and timely.
- The paper is very well written, with a detailed and mathematically solid presentation.
- I particularly like the idea behind Hista, which represents states using hidden embeddings and estimates rewards via nearest-neighbor reward aggregation. This approach is intuitive and conceptually appealing.
- The experimental setup is comprehensive and rigorous, covering multiple model scales (1.5B–14B) and a variety of tasks, including math reasoning, coding, science, and general QA.
- The results for Numca are especially impressive, showing improved mathematical reasoning performance and more stable training dynamics compared to GRPO and PPO.

# Weaknesses
- Confusing motivation in preliminary experiments (Section 4.2):
  - The motivation of the preliminary experiments on PPO in SVEB is somewhat confusing. My understanding is that Section 4.2 aims to demonstrate that critic value predictions are not sufficiently informative because they converge toward a simple average of group rewards.However, lines 114–126 (column 2) justify the use of Monte Carlo reference values as labels for training the critic. These MC reference values would usually be similar to the group average reward. If this is the case, then the critic converging toward the group-average reward would actually be expected behavior.
  - Furthermore, Figures 5–7 show that even with Hista, which is claimed to produce more accurate value estimates, the value distribution still largely converges toward the group-average reward. This raises two possible interpretations: Either Hista does not substantially improve the value estimation, or Convergence toward group-average rewards may not necessarily be problematic.
  - Clarifying this point would help strengthen the motivation of the proposed methods.
- Missing discussion of VAPO: An important related work that also addresses value-model-based RL for LLMs is VAPO. (cite https://arxiv.org/abs/2504.05118). Despite its relevance, VAPO is not mentioned in the related work section or included as a baseline. Given its conceptual overlap with this paper, it would be valuable for the authors to: discuss the relationship between VAPO and the proposed methods, clarify the conceptual differences, and provide empirical comparisons if possible.
- Presentation issues in the main results tables: Up until table 5, the authors highlight and underline the best metrics. However, this formatting is not applied consistently in Tables 5–7. Because these tables contain a large number of numerical results, the lack of highlighting makes them difficult and time-consuming to interpret. I recommend applying consistent formatting (e.g., highlighting best and second-best results) to improve readability.
- Limited performance gains for Hista: While Numca shows convincing improvements, the performance gains for Hista appear relatively modest. The proposed Hista algorithm is very intuitive and satisfying, and I also appreciate the authors for a very comprehensive and thorough experimental setup, but the performance gains are rather marginal and disappointing in my opinion. This makes it somewhat difficult to conclude that Hista provides a substantial practical benefit.
- Hista training efficiency: Hista requires extracting hidden states from the LLM during training. In my experience, this can significantly increase training time and GPU memory consumption. Given that the performance improvements appear incremental, it would be helpful to include at least a brief analysis of the additional computational overhead help readers assess whether the performance gains justify the additional computational cost.

---

> ### Author Rebuttal · Authors · 2026-03-31
>
> > ## Overall Response
>
> We thank the reviewer for the careful summary and constructive feedback. Below we clarify the main concerns and answer the questions.
>
> > ## Weakness 1.1: The motivation of preliminary experiments
>
> We apologize for the confusion caused by “group average” in the manuscript.
> 1. **In MCS (Monte Carlo Sampling):** for a given state $s_t$, we estimate its state value by conditioning on $s_t$, sampling $n$ future continuations, and averaging their terminal rewards. This is a **state-specific** reference value.
> 2. **In the PPO setting:** we freeze the actor and train only the critic. The critic collapses toward the group-average reward of the **initial state $s_0$**, the same baseline used in GRPO. As a result, different states within a trajectory receive nearly identical signals, which is the issue we highlight.
>
> > ## Weakness 1.2: Hista converges to group‑average reward
>
> We agree that Hista’s values may still center around the group average. However, Hista better captures the **tail** of the value distribution (Fig. 5–7), where deviations are large and crucial for fine-grained credit assignment. We show the distribution of absolute differences between group-average reward and MCS values on SVEB-MATH:
>
> | [-0.75,-0.3) | [-0.3,-0.2) | [-0.2, 0.1) | [0.1, 0) | [0, 0.1) | [0.1, 0.2) | [0.2,0.4) | [0.4, 0.9) |
> | ----------- | ----------- | ----------- | -------- | -------- | ---------- | --------- | ---------- |
> | 4.47        | 4.09        | 8.923       | 26.28    | 20.19    | 12.09      | 12.22      | 9.12       |
>
> Nearly 30% of the mass has an absolute difference larger than 0.2, so improving estimation in this region is meaningful even if the overall distribution remains centered.
>
> > ## Weakness 2: VAPO
>
> Thank you for pointing this out. We will add VAPO to the related work and discuss the connection more explicitly. Our focus is slightly different: VAPO studies critic behavior mainly from the perspective of value initialization bias, whereas our paper directly compares PPO predictions with ground-truth state values across training stages.
>
> > ## Weakness 4: Limited performance gain of Hista
>
> We view Hista mainly as a proof of concept showing that hidden-state similarity can help value estimation and credit assignment. We also provide more analysis on the modest improvement in the "Weakness 1" section of response to Reviewer Chui. In addition, our longer training run on R1-Distill-1.5B shows that the benefit can become larger with extended training with 3840 H800 hours.
>
> ### DAPO + Hista
>
> | steps	  |    0 | 4800 | 7200 | 12000 | 13200 | 14400 |
> | ------- | ---- | ---- | ---- | ----- | ----- | ----- |
> | AIME24  | 0.287| 0.304| 0.313| 0.321 | 0.346 | 0.404 |
>
> ### DAPO
>
> | steps	  |    0 | 4800 | 7200 | 12000 | 13200 | 14400 |
> | ------- | ---- | ---- | ---- | ----- | ----- | ----- |
> | AIME24  | 0.287| 0.303| 0.291| 0.316 | 0.356 | 0.305 |
>
> > ## Weakness 5: Training efficiency of Hista
>
> We monitor the average step time of the first 1200 training step of different models on MATH:
>
> |Avg Step Time (s)|Qwen2.5-14B-Base|Qwen3-4B-Base|Qwen3-1.7B-Base|
> | --------------- | -------------- | ----------- | ------------- |
> | DAPO            | 20.54          | 62.18       | 24.81         |
> | DAPO+Hista      | 21.07          | 62.82       | 25.01         |
>
> In most cases, Hista adds less than one second of latency per step, demonstrating its efficiency.
>
> > ## Key Question 1: Why Qwen2.5?
>
> Primarily because Qwen3 has both thinking and non-thinking mode, which might complicate our analysis. We also verified that Hista generalizes to Qwen3 base models:
>
> | Method | MATH | GSM | MINERVA | OLYMPIAD | AMC | AIME | GAOKAO | CollegeMath | Avg |
> | ---- | ---- | ---- | ---- | ---- | ---- | ---- | ---- | ---- | ---- |
> | 4B DAPO | 0.798 | 0.886 | 0.364 | 0.455 | 0.522 | 0.154 | 0.672 | 0.633 | 0.561 |
> | +Hista | 0.802 | 0.902 | 0.396 | 0.457 | 0.563 | 0.144 | 0.683 | 0.644 | 0.574 |
> | 8B DAPO | 0.830 | 0.933 | 0.402 | 0.518 | 0.588 | 0.202 | 0.740 | 0.648 | 0.607 |
> | +Hista | 0.860 | 0.933 | 0.413 | 0.550 | 0.584 | 0.225 | 0.743 | 0.664 | 0.622 |
>
> > ## Key Question 2: Training dynamics for Hista
>
> We will include the training curves for Hista in the revised appendix. The dynamics on Qwen2.5 Base are already shown in Figure 1 of the paper.
>
> > ## Key Question 3: Integration
>
> Both methods keep the original multi-sampling procedure. GRPO/DAPO still generate multiple rollouts per prompt and simply average the reward. Numca and Hista replace the baseline/advantage estimation step by hidden states similarity weighted average. Thus, they are plug-in replacements for the advantage calculation, not replacements for multi-sampling.
>
> > ## Key Question 4: Why not combine with PPO?
>
> Combining Hista with PPO is conceptually similar to combining it with GRPO, because Hista mainly changes the value/advantage estimation module. We therefore focused on GRPO and DAPO, where the effect is cleaner and more directly comparable.

---

> > ### Author Rebuttal · Reviewer_f7PS · 2026-04-01
> >
> > I thank the authors for the rebuttal. The rebuttal has cleared up and made the paper stronger. I increased the score to 5 for a clearer acceptance support.

---

> > > ### Author Response · Authors · 2026-04-03
> > >
> > > We thank the reviewer for their positive feedback and for increasing the score. We are pleased that our rebuttal addressed the concerns and clarified the contributions of our work. We will ensure that the additional clarifications and improvements discussed during the rebuttal are fully integrated into the final version of the manuscript.

---

### Official Review · Reviewer_Chui · 2026-03-15

**Soundness:** 2
**Presentation:** 2
**Significance:** 3
**Originality:** 3
**Overall Recommendation:** 4
**Confidence:** 4

**Summary:**

This paper studies state-value estimation for LLM RL, arguing that common group-average baselines and PPO critics fail to provide informative token-level guidance. The authors introduce SVEB, a benchmark that approximates reference state values with Monte Carlo continuations, then propose two estimators: Numca and Hista. The paper claims that Hista reduces value-estimation error across several SVEB domains and improves downstream RL training when added to methods such as DAPO and CSIPO.

**Compliance With Llm Reviewing Policy:**

Affirmed.

**Final Justification:**

The rebuttal has successfully resolved the main concerns I had.

**Key Questions For Authors:**

see weaknesses

**Limitations:**

yes

**Strengths And Weaknesses:**

## Strengths
- The paper targets a concrete and important weakness (credit assignment) in current LLM RL, and SVEB gives a useful lens for studying whether token-level value estimates are actually informative.
- Numca and Hista are conceptually simple and fit naturally into current RL pipelines without requiring extra learned critics or additional rollouts.
- The empirical section is fairly broad for this topic, covering multiple SVEB domains, several model sizes, and both benchmark-style evaluation and downstream RL training.

## Weaknesses
- The downstream gains are often modest and not uniformly convincing, with several tables showing mixed per-benchmark movement or only small average improvements, so the paper's consistent performance claims feel overstated.
- It is unclear whether the modest gains partly reflect the experimental setting itself: the paper studies sparse outcome-reward reasoning tasks, whereas the value-estimation problem may become more consequential in agentic RL settings with a clearer turn-level MDP.

---

> ### Author Rebuttal · Authors · 2026-03-31
>
> > ## Overall Response
>
> We thank the reviewer for the constructive feedback. We agree that the absolute gains reported in the original manuscript are sometimes modest. However, the main contribution of our work is not only improved final accuracy, but also showing that (i) verifiable milestones can accelerate convergence, and (ii) hidden states can serve as a general representation for such milestones. Since final RL performance depends on many factors, our method is best viewed as a plug-and-play value estimator that can be integrated into most RL algorithms and their variants.
>
> > ## Weakness 1: Performance consistency claim
>
> Our claim that Hista provides consistent gains is based on two observations: (1) Hista is compatible with multiple RL algorithms and improves performance on the majority of reasoning tasks; and (2) in our evaluated setting, it seldom degrade performance below the corresponding baseline.
>
> We also conducted additional experiments and analysis to better understand why the gains are sometimes modest.
>
> ### Possible reason 1: Evaluation settings
>
> To ensure efficiency, stability, and comparability with the base model, most benchmarks are evaluated with temperature = 0.0. For smaller benchmark sets such as AIME and AMC, we use temperature = 0.1 and report avg@16. In contrast, the training curves in Figure 1, where Hista shows consistent improvements over DAPO, are obtained with temperature = 0.7.
>
> To align training and evaluation settings, we re-evaluated all Qwen2.5 Base checkpoints trained on MATH with temperature = 0.7 and report avg@32, 95% CI, t-test p-values, and probability of superiority:
>
> | | Mean | 95% CI | p-value | Probability of Superiority |
> | ---- | ---- | ----- | ----- | ----- |
> | 0.5B DAPO | 0.179 | [0.168, 0.190] | - | - |
> | 0.5B DAPO+Hista | 0.198 | [0.186, 0.209] | 0.006 | 96.53 |
> | 1.5B DAPO | 0.347 | [0.334, 0.361] | - | - |
> | 1.5B DAPO+Hista | 0.361 | [0.346, 0.376] | 0.023 | 91.32 |
> | 1.5B CSIPO | 0.356 | [0.342, 0.372] | - | - |
> | 1.5B CSIPO+Hista | 0.360 | [0.345, 0.375] | 0.39 | 62.60 |
> | 3B DAPO | 0.423 | [0.405, 0.440] | - | - |
> | 3B DAPO+Hista | 0.434 | [0.413, 0.450] | 0.21 | 78.43 |
> | 3B CSIPO | 0.429 | [0.414, 0.445] | - | - |
> | 3B CSIPO+Hista | 0.445 | [0.428, 0.461] | 0.004 | 93.89 |
> | 7B DAPO | 0.529 | [0.514, 0.545] | - | - |
> | 7B DAPO+Hista | 0.541 | [0.525, 0.550] | 0.19 | 76.93 |
> | 7B CSIPO | 0.527 | [0.508, 0.545] | - | - |
> | 7B CSIPO+Hista | 0.540 | [0.521, 0.558] | 0.078 | 83.15 |
> | 14B DAPO | 0.544 | [0.530, 0.557] | - | - |
> | 14B DAPO+Hista | 0.558 | [0.542, 0.573] | 0.053 | 84.63 |
>
> After matching the training and evaluation settings, the improvement is more clearly visible. The additional statistical tests also support the robustness of our results.
>
> ### Possible reason 2: Missing credit
>
> Consider four disjoint trajectories:
> s0 → s1,1 → s1,2 → s1,3 → s_correct
> s0 → s2,1 → s2,2 → s2,3 → s_correct
> s0 → s3,1 → s3,2 → s3,3 → s_wrong
> s0 → s4,1 → s4,2 → s4,3 → s_wrong
>
> Assume that states from different trajectories are well separated in representation space. In this case, when estimating the value of an intermediate state (e.g., V(s1,1)), Hista primarily relies on continuations that pass through that specific state. Since all such continuations lead to correct outcomes, V(s1,1) will be close to 1.
>
> This creates a *concentration effect*: credit is assigned strongly to transitions that lead into such states (e.g., s0 → s1,1), while later transitions (e.g., s1,1 → s1,2) receive relatively less credit. As a result, Hista may emphasize a subset of “key” steps rather than distributing credit across the entire successful trajectory.
>
> In contrast, group-average reward methods assign the same advantage to all steps along a successful trajectory. While this is less precise, it ensures *full credit coverage*, meaning every step that contributes to success is reinforced.
>
> Therefore, when training is sensitivity to noisy gradients, Hista tends to perform better. When broader credit coverage is more important, the advantage over group-average methods may appear smaller. This helps explain why gains are sometimes modest despite improved value estimation accuracy.
>
> > ## Weakness 2: Experiment Setting
>
> We view this as a modeling issue. Although LLM reasoning is a sparse-reward problem, it can still be formulated as an MDP, with each generated token treated as one step. Under this view, state-value estimation is meaningful, and a better estimator can improve RL training. Therefore, we think the current setting is suitable for studying state-value estimation, though agentic RL may be an even more direct future application. We are also adding new results on Qwen3 and DeepSeek-distill models (Please refer to the "Weakness 4" and "Key Question 1" section in response to reviewer f7ps), which further support the generality of our method.

---

> > ### Author Rebuttal · Reviewer_Chui · 2026-04-04
> >
> > Thank you to the authors for the detailed rebuttal. My main concerns are resolved by the rebuttal. I decided to increase my score by 1.

---

### Official Review · Reviewer_K8ht · 2026-03-18

**Soundness:** 3
**Presentation:** 4
**Significance:** 3
**Originality:** 3
**Overall Recommendation:** 5
**Confidence:** 4

**Summary:**

This paper discusses accurate state value estimation during RL post-training. While methods such as PPO/GRPO rely on group average rewards, this paper discusses improving critic quality in LLM RL. Main contributions of this paper are SVEB benchmark, Numca and Hista.

**Compliance With Llm Reviewing Policy:**

Affirmed.

**Final Justification:**

Authors clarification/positioning of paper wrt phi-4 clarified non-uniform credit assignment across tokens. Previously i mistakenly assumed uniform credit assignment across tokens which might limit paper adoption.

**Key Questions For Authors:**

Q) Could you clarify if Theorem A.10 is referring to a weak correlation (or) strong bound especially in general case (A.3 A) ?

**Strengths And Weaknesses:**

Strengths:
- Good empirical analysis of limitation of current methods (PPO)
- Paper had strong analysis and math details

Potential weakness:
- Hista depends on richness of representation quality of hidden states
- Numca had weaker performance on science and general tasks highlighting lack of generalizability

---

> ### Author Rebuttal · Authors · 2026-03-31
>
> > ## Overall Response
>
> We thank the reviewer for the constructive feedback. Our work addresses the state value estimation gap in current RL (PPO/GRPO) by introducing fine-grained credit assignment. We use **Numca** to demonstrate the gains possible with verifiable milestones (e.g., in math), and **Hista** as a generalizable solution that leverages hidden-state similarity for advantage reweighting when explicit milestones are absent.
>
> > ## Weakness of Numca: Generalizability
>
> Regarding the reviewer’s concern about Numca’s generalizability, we wish to clarify that Numca was never intended as a general‑purpose solution. Instead, it serves as empirical evidence of the benefits of fine‑grained credit assignment, demonstrating “how much improvement can be gained“ when verifiable milestones are available. The underlying idea “identifying verifiable milestones” is itself generalizable; wherever such patterns exist in reasoning trajectories, Numca offers a simple yet effective approach to enhance RL training.
>
> > ## Weakness of Hista: Richness of Representation
>
> We agree Hista’s performance correlates with hidden-state richness. However, this is a natural property of LLMs following large-scale pretraining. To support this, we provide SVEB results across the Qwen2.5 and Qwen3 series.
>
> ### Qwen2.5-0.5B Instruct as actor
>
> | SVEB      | number | math  | science | general | programming |
> |-----------|--------|-------|---------|---------|-------------|
> | GRPO@40   | 0.134  | 0.153 | 0.195   | 0.120   | 0.160       |
> | HISTA@40  | 0.108  | 0.143 | 0.171   | 0.104   | 0.134       |
> | MCS@1     | 0.150  | 0.197 | 0.226   | 0.096   | 0.141       |
> | MCS@2     | 0.107  | 0.151 | 0.168   | 0.076   | 0.098       |
>
> ### Qwen2.5-1.5B Instruct as actor
>
> | SVEB      | number | math  | science | general | programming |
> |-----------|--------|-------|---------|---------|-------------|
> | GRPO@40   | 0.178  | 0.161 | 0.195   | 0.176   | 0.154       |
> | HISTA@40  | 0.148  | 0.139 | 0.172   | 0.162   | 0.118       |
> | MCS@1     | 0.194  | 0.236 | 0.225   | 0.155   | 0.176       |
> | MCS@2     | 0.146  | 0.156 | 0.168   | 0.124   | 0.120       |
>
> ### Qwen2.5-7B Instruct as actor
>
> | SVEB      | number | math  | science | general | programming |
> |-----------|--------|-------|---------|---------|-------------|
> | GRPO@40   | 0.212  | 0.176 | 0.196   | 0.162   | 0.142       |
> | HISTA@40  | 0.164  | 0.140 | 0.172   | 0.148   | 0.127       |
> | MCS@1     | 0.173  | 0.182 | 0.199   | 0.226   | 0.141       |
> | MCS@2     | 0.114  | 0.131 | 0.146   | 0.152   | 0.101       |
>
> ### Qwen3-0.6B Instruct as actor
>
> | SVEB | number | math | science | general | programming |
> | ---- | ---- | ---- | ---- | ---- | ---- |
> | GRPO@40 | 0.168 | 0.155 | 0.124 | 0.147 | 0.212 |
> | HISTA@40 | 0.159 | 0.151 | 0.115 | 0.140 | 0.204 |
> | MCS@1 | 0.267 | 0.281 | 0.186 | 0.186 | 0.208 |
> | MCS@2 | 0.180 | 0.193 | 0.130 | 0.129 | 0.142 |
>
> ### Qwen3-1.7B Instruct as actor
>
> | SVEB | number | math | science | general | programming |
> | ---- | ---- | ---- | ---- | ---- | ---- |
> | GRPO@40 | 0.175 | 0.136 | 0.125 | 0.137 | 0.154 |
> | HISTA@40 | 0.141 | 0.122 | 0.110 | 0.115 | 0.148 |
> | MCS@1 | 0.212 | 0.208 | 0.142 | 0.152 | 0.160 |
> | MCS@2 | 0.152 | 0.160 | 0.117 | 0.106 | 0.116 |
>
> ### Qwen3-4B Base as actor
>
> | SVEB | number | math | science | general | programming |
> | ---- | ---- | ---- | ---- | ---- | ---- |
> | GRPO@40 | 0.223 | 0.223 | 0.166 | 0.147 | 0.108 |
> | HISTA@40 | 0.146 | 0.139 | 0.154 | 0.107 | 0.088 |
> | MCS@1 | 0.196 | 0.190 | 0.107 | 0.136 | 0.052 |
> | MCS@2 | 0.139 | 0.140 | 0.084 | 0.080 | 0.058 |
>
> ### Qwen3-8B Base as actor
>
> | SVEB | number | math | science | general | programming |
> | ---- | ---- | ---- | ---- | ---- | ---- |
> | GRPO@40 | 0.144 | 0.173 | 0.226 | 0.164 | 0.135 |
> | HISTA@40 | 0.093 | 0.135 | 0.192 | 0.103 | 0.099 |
> | MCS@1 | 0.062 | 0.165 | 0.225 | 0.083 | 0.094 |
> | MCS@2 | 0.056 | 0.131 | 0.153 | 0.055 | 0.068 |
>
> Note: Lower SVEB scores indicate better estimation. Hista consistently outperforms GRPO across scales.
>
> > ## Question Regarding Theorem A.10
>
> Theorem A.10 refers to a weak correlation. While the formula in line 977 left column:
> $$
> P(R_1 = R_2) \geq \sum_{k=1}^V P_1(k)^2 - \max_k \|\mathbf{w}_{a,k}\|_2 \cdot D_L
> $$
>
> resembles a bound, one of the error term in $D_L$ contains
>
> $$
> \sum_{m=n+1}^\alpha \| \mathbf{x}_{1, m} - \mathbf{x}_{2,m} \|_2
> $$
>
> (the $L_2$ distance between hidden states of stochastically generated trajectories) is currently bounded by $2(\alpha - n)R_e$ to avoid over-complicating the analysis. In edge cases, a large $R_e$ makes the bound loose.
>
> Achieving a "strong" bound requires modeling the sequence in a specific high-dimensional space, which we are currently exploring. However, we believe this weak correlation is sufficient to motivate Hista, as evidenced by the improvements in both SVEB and actual training performance.

---

> > ### Author Rebuttal · Reviewer_K8ht · 2026-04-03
> >
> > Firstly I thank the authors for clarification
> >
> > Q) Curious how hista credit assignment (uniform v(s) for each token) compares at a high level to pivotal token concept in phi-4 paper (https://arxiv.org/pdf/2412.08905, Figure 3) ? (Curious about authors viewpoint)

---

> > > ### Author Response · Authors · 2026-04-03
> > >
> > > We thank the reviewer for the thoughtful follow-up and the reference to the Phi-4 work. Below, we clarify the token-level dynamics of Hista and provide a detailed comparison with Pivotal Token Steering (PTS).
> > >
> > > > ## Clarification on Hista's Credit Assignment
> > >
> > > We would like to clarify a potential misunderstanding regarding the term **"uniform"**. Hista does **not** assign a uniform $V^{\pi}(s)$ to each token. While our baseline (GRPO) assigns a uniform group-average reward to all tokens in a trajectory, Hista utilizes hidden state dynamics to estimate a distinct $V^{\pi}(s_t)$ for every token.
> > >
> > > > ## Comparative Analysis: Hista vs. Phi-4 (PTS)
> > >
> > > While both Hista and PTS share the motivation of moving beyond "coarse" trajectory-level rewards, they diverge significantly in their implementation and target tasks.
> > >
> > > > ### Conceptual Alignment (Motivation)
> > >
> > > The core insight of PTS in Phi-4 is that DPO assigns the same gradient weight to all tokens, failing to distinguish "pivotal" tokens that actually drive the outcome. Similarly, Hista addresses the "uniform advantage" issue in GRPO. In both cases, the goal is to identify tokens that significantly shift the model’s probability of success, $p(\text{success})$, which is equivalent to the state value $V^{\pi}(s_t)$ under our MDP modeling.
> > >
> > > > ### Methodological Divergence: Sampling vs. Representation
> > >
> > > The primary technical difference lies in how the "value" shift is measured:
> > >
> > > 1. **PTS (Phi-4):** Relies on **Monte Carlo Sampling (MCS)** to estimate $p(\text{success})$. By sampling multiple completions for a given token, they obtain a high-accuracy estimate of that token's importance. This is essentially the same "ground truth" estimation logic we use in our **SVEB**.
> > > 2. **Hista:** Instead of external sampling, Hista looks **inward**. We use the MD operation on hidden states to measure the distance between states to estimate state value. This allows us to estimate the "pivotal" nature of tokens without needing to generate a single additional completion.
> > >
> > > > ### Training Nature: Offline Alignment vs. Online Reasoning
> > >
> > > The choice of method is dictated by the training paradigm:
> > >
> > > 1. **Efficiency & Overhead:** PTS is applied in an **offline DPO** setting where data is pre-sampled. In this context, the high computational cost of MCS is an acceptable one-time "upfront" cost. However, in **online RL** (e.g., GRPO) for stimulating reasoning capability, performing MCS at every training step is prohibitively expensive (as noted in the DeepSeek-R1 report regarding MCTS in https://arxiv.org/abs/2501.12948 Appendix G2).
> > > 2. **Hista's Advantage:** Hista is designed specifically for online training. It achieves fine-grained credit assignment with **nearly zero speed or memory overhead** (please refer to "Weakness 5" section of reviewer f7ps and "Key Question 3" section of reviewer if7k for the efficiency comparison). This makes it a scalable solution for long-thought-chain reasoning models where multi-path sampling would be a bottleneck.
> > >
> > > > ### Experiments on the Overhead of MCS
> > >
> > > We introduce MCS to training by sampling 3 additional completions for every 50 tokens in trajectory and measure the average overhead in sampling stage:
> > >
> > > | Sampling Overhead (s) | Qwen3-8B-Base (8096 context) | Qwen3-4B-Base (8096 context) | R1-Distill-1.5B (16384 context) |
> > > | --------------------- | ---------------------------- | ---------------------------- | ------------------------------- |
> > > | MCS                   | 182                          |  226                         | 361                             |
> > >
> > > > ### Summary of Trade-offs
> > >
> > > | Dimension     |   PTS (Phi-4)                       | Hista (Ours)                      |
> > > | ------------- | ----------------------------------- | --------------------------------- |
> > > | Primary Goal  | Hallucination reduction / Alignment | Stimulating reasoning capabilities|
> > > | RL Framework  | Offline (DPO)                       | Online (GRPO)                     |
> > > | Value Estimation   | $p(\text{success})$ via sampling    | weighted average via hidden state distance |
> > > | Computational Cost | High (Multiple completions per token) | **Negligible** (Single forward pass) |
> > > | Implementation | Sampling-based            |     Representation-based         |
> > >
> > > > ## Conclusion
> > >
> > > In summary, PTS and Hista represent two different solutions to the credit assignment problem. PTS prioritizes maximum accuracy through expensive sampling for offline alignment. Hista prioritizes maximum efficiency through internal representation dynamics for online reasoning. **By recognizing the differing nature of these training objectives, one can apply the appropriate method according to the specific constraints of the task.** We will update our related work to discuss Phi-4 and contextualize Hista as a low-overhead alternative for online reasoning tasks.
> > >
> > > We thank the reviewer again for this insight, which helps us better position our work within the evolving landscape of RL.

---

### Decision · Program_Chairs · 2026-04-30

**Decision:**

Accept (spotlight)

**Comment:**

The paper studies and proposes a new techniques to improve accurate state value estimation in LLM reinforcement learning. The work is solid, with well-motivated presentation and writing. Experiment results are strong. Overall, all reviewers seem positive about this work. Authors did excellent job with addressing questions and concerns raised by reviewers in the rebuttal - for example, on computational overhead and statistical tests across various model scales. Overall, I recommend a strong accept.